# ALMOST LINEAR CONSTANT-FACTOR SKETCHING FOR $\ell_1$ AND LOGISTIC REGRESSION

**Alexander Munteanu & Simon Omlor**
Faculty of Statistics
TU Dortmund University
44227 Dortmund, Germany
`alexander.munteanu@tu-dortmund.de`
`simon.omlor@tu-dortmund.de`

**David P. Woodruff**
Department of Computer Science
Carnegie Mellon University
Pittsburgh, PA 15213, USA
`dwoodruf@cs.cmu.edu`

## ABSTRACT

We improve upon previous oblivious sketching and turnstile streaming results for $\ell_1$ and logistic regression, giving a much smaller sketching dimension achieving $O(1)$-approximation and yielding an efficient optimization problem in the sketch space. Namely, we achieve for any constant $c > 0$ a sketching dimension of $\tilde{O}(d^{1+c})$ for $\ell_1$ regression and $\tilde{O}(\mu d^{1+c})$ for logistic regression, where $\mu$ is a standard measure that captures the complexity of compressing the data. For $\ell_1$-regression our sketching dimension is near-linear and improves previous work which either required $\Omega(\log d)$-approximation with this sketching dimension, or required a larger $\mathrm{poly}(d)$ number of rows. Similarly, for logistic regression previous work had worse $\mathrm{poly}(\mu d)$ factors in its sketching dimension. We also give a tradeoff that yields a $1 + \varepsilon$ approximation in input sparsity time by increasing the total size to $(d \log(n)/\varepsilon)^{O(1/\varepsilon)}$ for $\ell_1$ and to $(\mu d \log(n)/\varepsilon)^{O(1/\varepsilon)}$ for logistic regression. Finally, we show that our sketch can be extended to approximate a regularized version of logistic regression where the *data-dependent* regularizer corresponds to the variance of the individual logistic losses.

## 1 INTRODUCTION

We consider logistic regression in distributed and streaming environments. A key tool for solving these problems is a distribution over random oblivious linear maps $S \in \mathbb{R}^{r \times n}$ which have the property that, for a given $n \times d$ matrix $X$, where we assume the labels for the rows of $X$ have been multiplied into $X$, given only $SX$ one can efficiently and approximately solve the logistic regression problem. The fact that $S$ does not depend on $X$ is what is referred to as $S$ being oblivious, which is important in distributed and streaming tasks since one can choose $S$ without first needing to read the input data. The fact that $S$ is a linear map is also important for such tasks, since given $SX^{(1)}$ and $SX^{(2)}$, one can add these to obtain $S(X^{(1)} + X^{(2)})$, which allows for positive or negative updates to entries of the input in a stream, or across multiple servers in the arbitrary partition model of communication, see, e.g., (Woodruff, 2014) for a discussion of data stream and communication models.

An important goal is to minimize the *sketching dimension* $r$ of the sketching matrix $S$, as this translates into the memory required of a streaming algorithm and the communication cost of a distributed algorithm. At the same time, one would like the approximation factor that one obtains via this approach to be as small as possible. Specifically we develop and improve oblivious sketching for the most important *robust* linear regression variant, namely $\ell_1$ regression, and for logistic regression, which is a *generalized* linear model of high importance for binary classification and estimation of Bernoulli probabilities. Sketching supports very fast updates which is desirable for performing robust and generalized regression in high-velocity data processing applications, for instance in physical experiments and other resource constraint settings, cf. (Munteanu et al., 2021; Munteanu, 2023).

We focus on the case where the number $n$ of data points is very large, i.e., $n \gg d$. In this case, applying a standard algorithm directly is not a viable option since it is either too slow or even becomes impossible when it requires more memory than we can afford. Following the *sketch & solve* paradigm

(Woodruff, 2014), our goal is in a first step to reduce the size of the data without losing too much information. Then, in a second step, we approximate the problem efficiently on the reduced data.

**Sketch & solve principle:**
1. Calculate a small sketch $SX$ of the data $X$.
2. Solve the problem $\tilde{\beta} = \operatorname{argmin}_\beta f(SX\beta)$ using a standard optimization algorithm.

The theoretical analysis proves that the sketch in the first step is calculated in such a way that the solution obtained in the second step is a good approximation to the original problem, i.e., that $f(X\tilde{\beta}) \leq C \cdot \operatorname{argmin}_\beta f(X\beta)$ holds for a small constant factor $C \geq 1$.

## 1.1 OUR CONTRIBUTIONS

For logistic regression our goal is to achieve an $O(1)$-approximation with an efficient estimator in the sketch space and smallest possible sketching dimension in terms of $\mu$ and $d$, where $\mu = \mu(X) = \sup_{\beta \neq 0} \frac{\sum_{x_i\beta > 0} |x_i\beta|}{\sum_{x_i\beta < 0} |x_i\beta|}$ is a data dependent parameter that captures the complexity of compressing the data for logistic regression, see Definition 2.1. As a byproduct of our algorithms, we also obtain algorithms for $\ell_1$-regression. We note that the parameter $\mu$ is necessary only for logistic regression, i.e., for sketching $\ell_1$-regression, we set $\mu = 1$. We summarize our contributions as follows:

1) We significantly improve the sketch of Munteanu et al. (2021). More precisely we show with minor modifications in their algorithm but major modifications in the analysis that the size of the sketch can be reduced from roughly $\tilde{O}(\mu^7 d^5)$[1] to $\tilde{O}(\mu d^{1+c})$ for any $c > 0$, while preserving an $O(1)$ approximation to either the logistic or $\ell_1$ loss.

2) We show that increasing the sketching dimension to $(\mu d \log(n)/\varepsilon)^{O(1/\varepsilon)}$ is sufficient to obtain a $1 + \varepsilon$ approximation guarantee.

3) We show that our sketch can also approximate variance-based regularized logistic regression within an $O(1)$ factor if the dependence on $n$ in the sketching dimension is increased to $n^{0.5+c}$ for any $c > 0$. We also give an example corroborating that the `CountMin`-sketch that we use needs at least $\Omega(n^{0.5})$ rows to achieve an approximation guarantee below $\log^2(\mu)$.

## 1.2 RELATED WORK

**Data oblivious sketching** Data oblivious sketches have been developed for many problems in computer science, see (Phillips, 2017; Munteanu, 2023) for surveys. The seminal work of Sarlós (2006) opened up the toolbox of sketching for numerical linear algebra and machine learning problems, such as linear regression and low rank approximation, cf. (Woodruff, 2014). We note that oblivious sketching is very important to obtain data stream algorithms in the turnstile model (Muthukrishnan, 2005) and there is evidence that linear sketches are optimal for such algorithms under certain conditions (Li et al., 2014; Ai et al., 2016). The classic works on $\ell_2$ regression have been generalized to other $\ell_p$ norms (Sohler & Woodruff, 2011; Woodruff & Zhang, 2013) by combining sketching as a fast but inaccurate preconditioner and subsequent sampling to achieve the desired $(1 + \varepsilon)$-approximation bounds. Those works have been generalized further to so-called $M$-estimators, i.e., Huber (Clarkson & Woodruff, 2015a) or Tukey regression loss (Clarkson et al., 2019), that share nice properties such as symmetry and homogeneity leveraged in previous works on $\ell_p$ norms.

$\ell_1$ **regression** Specifically for $\ell_1$, the first sketching algorithms used random variables drawn from 1-stable (Cauchy) distributions to estimate the norm (Indyk, 2006). It is possible to get concentration and a $(1 \pm \varepsilon)$-approximation in near-linear space by using a median estimator. However, in a regression setting this estimator leads to a non-convex optimization problem in the sketch space. Since we want to preserve convexity to facilitate efficient optimization in the sketch space, we focus on sketches that work with an $\ell_1$ estimator for solving the $\ell_1$ regression problem in the sketch space in order to obtain a constant approximation for the original $\ell_1$ problem. With this restriction, it is possible to obtain a contraction bound with high probability so as to union bound over a net, but similar results are not available for the dilation. Indeed, *subspace embeddings* for the $\ell_1$ norm have

---

[1]The tilde notation suppresses *any* $\operatorname{polylog}(\frac{\mu dn}{\varepsilon\delta})$ even if no higher order terms appear. This allows us to focus on the main parameters and their improvement. The exact terms are specified in Theorems 1-3.

$\tilde{\Theta}(d)$ dilation (Woodruff & Zhang, 2013; Li et al., 2021; Wang & Woodruff, 2022). A $1+\varepsilon$ dilation is only known to be possible when mapping to $\exp(O(1/\varepsilon))$ dimensions (Brinkman & Charikar, 2005), even for single vectors as in (Indyk, 2006). We thus focus on obtaining an $O(1)$ approximation in this paper. Previous work had either larger $O(\log(d))$ distortion[2] or larger $\text{poly}(d)$ factors (Indyk, 2006; Sohler & Woodruff, 2011). There exists a $(1 + \varepsilon)$-approximation algorithm (Sohler & Woodruff, 2011) for turnstile data streams, running two sketches in parallel: one for preconditioning and another that performs $\ell_1$-row-sampling from the *sketch* (Andoni et al., 2009). However, it has a worse $\text{poly}(d\log(n)/\varepsilon)$ update time and sketching dimension, see (Sohler & Woodruff, 2011, Theorem 13). An advantage of our sketch is that it uses only random $\{0,1\}$-entries, which have better computational and implicit storage properties (Alon et al., 1986; 1999; Rusu & Dobra, 2007). More importantly, our approach works simultaneously for both, $\ell_1$ *and* logistic regression. For the latter no near-linear sketching dimension was known to be possible since sketches for $\ell_1$ cannot preserve the sign of coordinates, which is crucial for any multiplicative error on the asymmetric logistic loss.

**Generalized linear models (GLMs)** It is important to extend the works on linear regression to more sophisticated and expressive statistical learning problems, such as *generalized linear models* (McCullagh & Nelder, 1989). Unfortunately, taking this step led to impossibility results. Namely, approximating the regression problems on a succinct sketch for strictly monotonic functions such as logistic loss (Munteanu et al., 2018) or heavily imbalanced asymmetric functions such as Poisson regression loss (Molina et al., 2018) allows one to design a low-communication protocol for the INDEXING problem that contradicts its $\Omega(n)$ bit randomized one-way communication complexity (Kremer et al., 1999). This implies an $\tilde{\Omega}(n)$ sketching dimension for these problems. To circumvent this worst-case limitation for logistic regression, Munteanu et al. (2018) introduced a natural data dependent parameter $\mu$ that can be used to bound the complexity of compressing data for logistic and probit regression (Munteanu et al., 2022). This also led to the very first oblivious sketch for logistic regression (Munteanu et al., 2021), with a polylogarithmic number of rows for mild data. We improve this by giving, the *only* near-linear sketching dimension in $d$ and $\mu$ for logistic regression. The previous best sketching dimension obtained by Lewis weight sampling (Mai et al., 2021), required $O(\mu^2 d)$ and crucially their sketch is not oblivious so cannot be implemented in a turnstile data stream, with positive and negative updates to the entries of the input point set. For lower bounds, an $\Omega(d)$ dependence is immediate since mapping to fewer than $d$ dimensions contracts non-zero vectors in the null-space of the sketching matrix to zero. An $\Omega(\mu)$ lower bound is immediate from Munteanu et al. (2018) and was recently generalized by Woodruff & Yasuda (2023) to more natural settings.

**Variance-based regularization** Regularization techniques have been proposed in the literature for many purposes, such as reducing the effective dimension of statistical problems or limiting their expressivity to avoid overfitting. Regularization was also proposed to relax the logistic regression problem. In an extreme setting where the regularizer dominates the objective function, the contributions of data points do not differ significantly. The problem then becomes easy to approximate by uniform subsampling (Samadian et al., 2020). To address the bias-variance tradeoff in machine learning problems in a more meaningful way and to provably reduce the generalization error of models, Maurer & Pontil (2009) proposed to add a *data-dependent* variance-based regularization. Since this results in a non-convex optimization problem even for convex objectives, Duchi & Namkoong (2019); Yan et al. (2020) used optimization tricks to reformulate a convex variant with additional parameters that can be integrated into standard hyperparameter tuning. Interestingly, this *data-dependent* regularization – in contrast to standard regularization – does not relax the sketching problem but makes it more complicated, requiring in the case of logistic regression a combination of $\ell_1$ and $\ell_2$ geometries to be preserved. We show that our sketch can deal with both simultaneously.

## 1.3 OUR TECHNIQUES

Our main motivation is to reduce the large dependence on the parameters of the oblivious sketching algorithm of Munteanu et al. (2021). Their sketch consists of $O(\log n)$ levels that take subsamples at exponentially decreasing rate, and apply a `CountMin`-sketch to each subsample to compress it to roughly size $\tilde{O}(d^5(\mu/\varepsilon)^7)$ which gives $(1-\varepsilon)$ contraction but only $O(1)$[3] dilation bounds. Our new methods significantly improve over their sketching dimension for obtaining $(1 - \varepsilon)$ contraction bounds. The large dependence on $\mu$ came from adapting the analysis of Clarkson & Woodruff (2015a)

---

[2]by an argument in the proof of Lemma 7 of Sohler & Woodruff (2011), cf. (Woodruff, 2021, Problem 1).

[3]The exact constant was not specified but overcounting their parameters gives at best an $\geq 8$-approximation.

to work for the asymmetric logistic loss function. This required to rescale $\varepsilon' = \varepsilon/\mu$ to translate the estimation error of $\varepsilon'\|z\|_1$ to an error of $\varepsilon\|z^+\|_1$, where the latter quantity sums only over the positive entries of $z$. We avoid this by noting that we can oversample the elements by a factor of $\mu$ to capture sufficiently many elements to approximate the required $\varepsilon\|z^+\|_1$ error directly. However, the analysis of the so-called *heavy hitters*[4] requires $\mu$ elements to be perfectly isolated when hashing them into buckets, which requires $\mu^2$ buckets to succeed with good probability. To obtain a linear dependence on $\mu$, we sacrifice some sparsity in our sketching matrix. Instead of hashing to a single bucket at each level, we hash each element multiple times. The best known trade-off between sketching dimension and sparsity is due to Cohen (2016) for the `Count`-sketch. We adapt the technique to our `CountMin`-sketch: we hash each element to roughly $O(\varepsilon^{-1}\log(\mu d/\varepsilon))$ buckets and resolve collisions by summing the elements instead of taking a random sign combination. This brings the dependence on $\mu$ down to quasi-linear. The dependence on $d$ and $\varepsilon$ also benefit from this technique but at this point, our analysis still requires a $d^2$ dependence. This comes from needing to separate the heavy hitters from other large coordinates for all vectors in a net of exponential size in $d$. To bring the dependence on $d$ down to near-linear we densify our sketch to roughly $O(\varepsilon^{-3}\mu d\log(n))$ non-zeros per column, which separates the heavy hitters almost entirely and yields our result.

We also improve the dilation bounds from a factor of $\geq 8$ to a 2-approximation: previous analyses were conducted by bounding the expected contribution of weight classes $W_q = \{i \mid 2^{-q-1} < z_i \leq 2^{-q}\}$ to the different levels in our sketch. A simple bound of $O(\log n)$ was improved to $O(1)$ by a Ky-Fan norm argument, which cuts off elements that have a low redundant contribution. We reverse the perspective and ask for each level $h$, which weight classes are well represented? This allows us to conduct a more fine-grained analysis: we define intervals $Q_h = [q(2), q(3)]$ and $Q_h \subseteq Q'_h = [q(1), q(4)]$ and quantify their size depending on the number of buckets, such that weight classes $q \notin Q'_h$ do not contribute at all, $q \in Q'_h$ make a non-negligible contribution, and $q \in Q_h$ are additionally well-approximated. It is thus desirable to choose the number of buckets in such a way that $|Q'_h|/|Q_h| \approx 1$. Moreover, we ask how the intervals in consecutive levels overlap. It turns out that slightly increasing the number of buckets in each level to $\tilde{O}(\mu d^2)$ allows us to show that each $q$ appears only in at most two consecutive levels (in expectation) which yields a 2-approximation. Indeed, the argument can be continued by raising the size of the sketch to any power of $k \in \mathbb{N}$, resulting in an expected contribution in at most $(1 + 1/k)$ levels, which yields a $(1 + \varepsilon)$-approximation using $(\mu d\log(n)/\varepsilon)^{O(1/\varepsilon)}$ rows. An exponential dependence on $1/\varepsilon$ is best known for sketching-based estimators of the $\ell_1$ norm (Indyk, 2006; Li et al., 2021) that embed into lower-dimensional $\ell_1$, and our sketch can be used as such an estimator for $\ell_1$ as a special case.

Finally, as a corollary and important application of our results, we obtain similar oblivious sketching bounds for a variance-regularized version of logistic regression, see Section 1.2. It combines aspects of the $\ell_1$ geometry of the sum of logistic losses with the $\ell_2$ geometry that appears in the sum of *squared* logistic losses. The analysis is very similar to the standard logistic regression loss but requires redefining the weight classes in terms of squared values $z_i^2$ and converting between the two norms, which introduces roughly another $O(\sqrt{n})$-factor. Previous work on data reduction methods for generalized linear models that work for different $\ell_p$ losses were either based on sampling (Munteanu et al., 2022) or worked only for symmetric functions such as norms (Clarkson & Woodruff, 2015a). However, an oblivious sketch for our loss function requires preserving the signs of elements which is not possible with previous sketching methods. Relying on the `CountMin`-sketch as in (Munteanu et al., 2021) thus seems necessary. For this choice we show that an additional $\Theta(\sqrt{n})$-factor is unavoidable and hereby we corroborate the tightness of our analysis.

## 2 PRELIMINARIES AND MAIN RESULTS

For $\ell_1$ regression, we consider as inputs a data matrix $X \in \mathbb{R}^{n \times d}$ and a target vector $Y \in \mathbb{R}^n$. The task is to find $\beta \in \operatorname{argmin}_{\beta \in \mathbb{R}^d} \|X\beta - Y\|_1$. We note that up to constants, this corresponds to minimizing the negative log-likelihood of a standard linear model $Y = X\beta + \eta$ with a Laplace noise distribution $\eta_i \sim L(0, 1)$ for all $i \in [n]$. Our goal will be to design an oblivious linear sketching matrix $S$ such that the sketch $X' = S[X, Y]$ is significantly reduced in its number of rows and solving the compressed $\ell_1$ regression problem in the sketch space yields an $O(1)$ approximation to the same problem on the original large data.

---

[4]i.e., the coordinates of $z$ with largest $\ell_1$ leverage score

Up to slight modifications, this sketch will also allow us to approximate logistic regression within a constant factor. For logistic regression, assume that we are given a data set $Z = \{z_1, \ldots, z_n\}$ with $z_i \in \mathbb{R}^d$ for all $i \in [n]$, together with a set of labels $Y = \{y_1, \ldots, y_n\}$ with $y_i \in \{-1, 1\}$ for all $i \in [n]$. In logistic regression the negative log-likelihood (McCullagh & Nelder, 1989) is of the form

$$\mathcal{L}(\beta | Z, Y) = \sum_{i=1}^n \ln(1 + \exp(-y_i z_i \beta)),$$

which, from a learning and optimization perspective, is the objective function that we would like to minimize. For $r \in \mathbb{R}$ we set $\ell(r) = \ln(1 + \exp(r))$ to simplify notation. Then we have that $\mathcal{L}(\beta | Z, Y) = \sum_{i=1}^n \ell(-y_i z_i \beta)$. We also include a variance-based regularization as proposed in (Maurer & Pontil, 2009; Duchi & Namkoong, 2019; Yan et al., 2020) to decrease the generalization error. We view our data set as $n$ realizations of a random variable $(z, y)$, where each $(z_i, y_i)$ is drawn i.i.d. from an unknown distribution $\mathcal{D}$. Then the expected value of the negative log-likelihood (on the empirical sample) for any fixed $\beta$ equals $\mathbb{E}(\ell(-yz\beta)) = \frac{1}{n} \mathcal{L}(\beta | Z, Y)$. The variance is given by

$$\mathrm{Var}(\ell(-yz\beta)) = \mathbb{E}(\ell(-yz\beta)^2) - \mathbb{E}(\ell(-yz\beta))^2 = \frac{1}{n} \sum_{i=1}^n \ell(-y_i z_i \beta)^2 - \left(\frac{1}{n} \sum_{i=1}^n \ell(-y_i z_i \beta)\right)^2$$

We also introduce a regularization hyperparameter $\lambda \in \mathbb{R}_{\geq 0}$. Then our objective is to minimize

$$\mathbb{E}(\ell(-yz\beta)) + \frac{\lambda}{2} \mathrm{Var}(\ell(-yz\beta)).$$

As $z_i$ and $y_i$ always appear together, we set $x_i = -y_i z_i$. Further we set $X \in \mathbb{R}^{n \times d}$ to be the matrix with row vectors $x_i$ for $i \in [n]$. For technical reasons we include a weight vector $w \in \mathbb{R}_{\geq 0}^n$ into the objective. Then our goal is to find $\beta \in \mathbb{R}^d$ minimizing

$$f_w(X\beta) = \frac{1}{n} \sum_{i=1}^n w_i \ell(x_i \beta) + \frac{\lambda}{2n} \sum_{i=1}^n w_i \ell(x_i \beta)^2 - \frac{\lambda}{2} \left(\frac{1}{n} \sum_{i=1}^n w_i \ell(x_i \beta)\right)^2.$$

The unweighted case corresponds to choosing $w$ to be the vector containing only 1's, in which case we set $f(X\beta) = f_w(X\beta)$.

We also note that $f(X\beta) \geq \frac{1}{n} \sum_{i=1}^n \ell(x_i \beta)$ since the variance term is non-negative. Next, observe that $\min_{\beta \in \mathbb{R}^d} f(X\beta) \leq f(0) = \ell(0) = \ln(2)$. In our analysis we investigate functions $\ell(r)$ and $\ell(r)^2$. Further we split $f$ into three functions $f_1(X\beta) = \frac{1}{n} \sum_{i=1}^n \ell(x_i \beta)$, $f_2(X\beta) = \frac{\lambda}{2n} \sum_{i=1}^n \ell(x_i \beta)^2$, and $f_3(X\beta) = \frac{\lambda}{2} \left(\frac{1}{n} \sum_{i=1}^n \ell(x_i \beta)\right)^2 = \frac{\lambda}{2} f_1(X\beta)^2$.

In contrast to the $\ell_1$ regression problem, a data reduction for $f$ or even $f_1$ where the sketch size is $r \ll n$ cannot be obtained in general. There are examples where no sketch of size $r = o(n/\log n)$ exists, even for an arbitrarily *large* but finite error bound (Munteanu et al., 2018). If we require the sketch to be a subset of the input, the bound can be strengthened to $\Omega(n)$ (Tolochinsky et al., 2022). Those impossibility results rely on the monotonicity of the loss function and thus extend to the function $f$ studied in this paper. To get around these strong limitations, Munteanu et al. (2018) introduced a parameter $\mu$ as a natural notion for parameterizing the complexity of compressing the input matrix $X$ for logistic regression. It was recently adapted for $p$-generalized probit regression (Munteanu et al., 2022). We work with a similar generalization given in the following definition.

**Definition 2.1.** *Let $X \in \mathbb{R}^{n \times d}$ be any matrix and let $p \in [1, \infty)$. We define*

$$\mu_p(X) = \sup_{\beta \in \mathbb{R}^d \setminus \{0\}} \frac{\sum_{x_i \beta > 0} |x_i \beta|^p}{\sum_{x_i \beta < 0} |x_i \beta|^p}.$$

*We say that $X$ is $\mu$-complex if $\max\{\mu_1(X), \mu_2(X)\} \leq \mu$.*

Our goal is to construct a slightly relaxed version of a sketch that suffices to obtain a good approximation by optimizing in the sketch space:

**Definition 2.2.** *Given a dataset $(X, w)$, a subset $V \subset \mathbb{R}^d$, $a > 1$ and $\varepsilon, \delta > 0$. A weak weighted $(V, a, \varepsilon)$-sketch $C = (X', w')$ for $f$ is a matrix $X' \in \mathbb{R}^{r \times d}$ together with a weight vector $w' \in \mathbb{R}_{>0}^r$ such that it holds simultaneously that: For all $\beta \in V$ we have*

$$f_{w'}(X'\beta) \geq (1 - \varepsilon) f_w(X\beta)$$

*and for $\beta^* \in V$ minimizing $f_w(X\beta)$ it holds that*

$$f_{w'}(X'\beta^*) \leq a f_w(X\beta^*).$$

*Further for any $\beta \in \mathbb{R}^d \setminus V$ it holds that*

$$f_{w'}(X'\beta) > \min_{\beta \in V} f_{w'}(X'\beta).$$

We note that a sketch satisfying the definition for $V = \mathbb{R}^d$ is known in the literature as a *lopsided embedding* (Sohler & Woodruff, 2011; Clarkson & Woodruff, 2015b; Feng et al., 2021). We denote by $\mathtt{nnz}(X)$ the number of non-zero entries of $X$. Our main results are the following:

For $\ell_1$ regression, where the objective is $\|X\beta - Y\|_1$, we have

**Theorem 1.** *Let $X \in \mathbb{R}^{n \times d}$ and let $Y \in \mathbb{R}^n$. Let $\varepsilon, \delta > 0$ and let $a > 1$. Then there is a distribution over sketching matrices $S \in \mathbb{R}^{r \times n}$ and a corresponding weight vector $w \in \mathbb{R}^r$, for which $X' = S[X, Y]$ can be computed in $T$ time in a single pass over a turnstile data stream such that $(X', w)$ is a weak weighted $(\mathbb{R}^d, \alpha, \varepsilon)$-sketch for $\ell_1$-regression with failure probability at most $P$, where*

    *1. $r = O(d^{1+c} \ln(n)^{3+5c})$ for any constant $1 \geq c > 0$, $T = O(d \ln(n)\mathtt{nnz}(X))$, and $\alpha = 1 + \frac{1}{c}$ and $P$ are constant,*

    *2. $r = O(\frac{d^4 \ln(n)^5}{\delta \varepsilon^7}) + \frac{32d \ln(n)^3}{\varepsilon^5} \cdot (\frac{64d \ln(n)^5}{\varepsilon^6 \delta})^{1+\varepsilon^{-1}}$, $T = O(\mathtt{nnz}(X))$, $\alpha = (1 + a\varepsilon)$, and $P = \delta + \frac{1}{a}$.*

For logistic regression where the objective function is only $f_1(X\beta)$, we have

**Theorem 2.** *Let $1 \geq c > 0$ be any constant. Let $X \in \mathbb{R}^{n \times d}$ be a $\mu$-complex matrix for bounded $\mu \in O((d \log^3(n))^c)$. Let $\varepsilon, \delta > 0$ and let $a > 1$. Then there is a distribution over sketching matrices $S \in \mathbb{R}^{r \times n}$ and a corresponding weight vector $w \in \mathbb{R}^r$, for which $X' = SX$ can be computed in $T$ time in a single pass over a turnstile data stream such that $(X', w)$ is a weak weighted $(\mathbb{R}^d, \alpha, \varepsilon)$-sketch for $f_1$ with failure probability at most $P$, where*

    *1. $r = O(\mu d^{1+c} \ln(n)^{2+4c})$, $T = O(\mu d \ln(n)\mathtt{nnz}(X))$, and $\alpha = 1 + \frac{1}{c}$ and $P$ are constant,*

    *2. $r = O(\frac{d^4 \ln(n)^5 \mu^2}{\delta \varepsilon^7}) + \frac{32d\mu \ln(n)^2}{\varepsilon^5} \cdot (\frac{64d \ln(n)^4}{\varepsilon^7 \delta})^{1+\varepsilon^{-1}}$, $T = O(\mathtt{nnz}(X))$, $\alpha = (1 + a\varepsilon)$, and $P = \delta + \frac{1}{a}$.*

Note that setting $a = \delta^{-1}$ and substituting $\varepsilon$ with $\varepsilon\delta$ in the second item yields a $(1+\varepsilon)$ approximation with probability at least $1 - 2\delta$.

For the variance-based regularization, where we consider the full objective function $f(X\beta)$, we have

**Theorem 3.** *Let $X \in \mathbb{R}^{n \times d}$ be a $\mu$-complex matrix for bounded $\mu < n$. Let $\varepsilon, \delta > 0$, let $a > 1$ and set $V = \{X\beta \mid f_1(X\beta) \leq \ln(2)(1 - \varepsilon)\}$. Then there is a distribution over sketching matrices $S \in \mathbb{R}^{r \times n}$ and a corresponding weight vector $w \in \mathbb{R}^r$, for which $X' = SX$ can be computed in $T$ time in a single pass over a turnstile data stream such that $(X', w)$ is a weak weighted $(V, \alpha, \varepsilon)$-sketch for $f$ with failure probability at most $P$, where*

    • *$r = O(\frac{n^{0.5+c} \mu d^2 \ln^3(n)}{\varepsilon^5} \cdot \max\{d, \ln(n), \varepsilon^{-1}, \delta^{-1}, \mu\} + \frac{d^5 \mu^2 \ln(n)^5 \sqrt{n}}{\delta \varepsilon^7})$, for arbitrary constant $1 \geq c > 0$, $T = O(\mathtt{nnz}(X))$, $\alpha = 1 + \frac{a}{c}$, and $P = \delta + \frac{1}{a}$.*

We note that for generality of our results we specify a tradeoff between our $d^{1+c}$ dependence and an arbitrarily large constant approximation error $\alpha = 1 + 1/c$. We stress that specific parameterizations yield strictly improved results over previous work. For instance we improve the $\geq 8$-approximation of (Munteanu et al., 2021) within $\tilde{O}(\mu^7 d^5)$ to a 2-approximation within $\tilde{O}(\mu d^2)$ by choosing $c = 1$. We further improve several lopsided $\ell_1 \to \ell_1$ embedding results to a factor of 2 where previous approximations gave only $O(d \log d)$ (Sohler & Woodruff, 2011), $O(\log d)$ (Woodruff, 2021), or $\geq 8$ (Clarkson & Woodruff, 2015a) or gave only non-convex estimators in the sketch space (Backurs et al., 2016) along with larger superlinear dependencies on $d$.

**Technical description of the sketch** Our contributions lie mainly in the improved and refined theoretical analyses. The sketching matrices of Theorems 1-3 are the same as in (Munteanu et al., 2021) up to small but important algorithmic modifications specified in the textual description below, and in pseudo-code, see Algorithm 1 in the appendix. The sketching matrix consists of $O(\log n)$ levels. In each level we take a subsample of all rows $i \in [n]$ at a different rate and hash the sampled items uniformly to a small number of buckets. All items that are mapped to the same bucket are summed up. This corresponds to a `CountMin` sketch (Cormode & Muthukrishnan, 2005) applied to the subsample taken at each level. More specifically, we will use the following parameters:

- $h_m$: the number of levels,
- $N_h$: the number of buckets at level $h$,
- $p_h$: the probability that any element $x_i$ is sampled at level $h$.

As we read the input, we sample each element $x_i$ for each level $h \leq h_m$ with probability $p_h$. The sampling probabilities are exponentially decreasing, i.e., $p_h \propto 1/b^h$ for some $b \in \mathbb{R}$ with $b > 1$. The weight of any bucket at level $h$ is set to $1/p_h$. At level $h_m$, we have $p_{h_m} \propto \frac{1}{n}$. It thus corresponds to a small uniform subsample and the number of buckets is equal to the number of rows that are sampled, i.e., $N_u := N_{h_m} \approx np_{h_m} =: np_u$. At level 0 we sample all rows, i.e., $p_0 = 1$ and the number of buckets is either the same as for the levels $h \in (0, h_m)$ or less. Consequently level 0 is a standard CountMin sketch of the entire data. All levels $h \in (0, h_m)$ have the same number of buckets $N_h = N$. For obtaining subquadratic dependence on $\mu, d$ in item 1 of Theorems 1 and 2, the sketch at level 0 is densified, which means that each element is hashed to a number of $s > 1$ buckets. The idea of the sketching algorithm is that for each fixed $\beta \in \mathbb{R}^d$ we partition the coordinates of $X\beta$ into weight classes depending on their contribution to the objective function. Each level approximates well a certain range of weight classes if their total contribution is large enough. For example the highest level $h_m$ will cover all the elements in small weight classes and the lowest level 0 will capture the so-called *heavy hitters* that appear rarely but have a significant contribution to the objective. Another algorithmic change is randomizing the size of the sketch at level 0, which is crucial for obtaining a $(1 + \varepsilon)$-approximation. For the exact details we refer to the analysis and Assumption A.1.

## 3 HIGH LEVEL DESCRIPTION OF OUR NOVEL ANALYSIS

Several details of the analysis, such as assumptions on the parameters, technical lemmas, and proofs are deferred to the appendix. We start by splitting the functions $f_1$ and $f_2$ into multiple parts:

**Lemma 3.1.** *It holds that* $nf_1(X\beta) = \sum_{x_i\beta>0} |x_i\beta| + \sum_{i=1}^n \ell(-|x_i\beta|)$ *and similarly we have that* $nf_2(X\beta) = \sum_{x_i\beta>0} |x_i\beta|^2 + 2\sum_{x_i\beta>0} \ell(-|x_i\beta|) \cdot |x_i\beta| + \sum_{i=1}^n \ell(-|x_i\beta|)^2$.

This can be used in the following way: if all $x_i\beta$ make only small contributions then uniform sampling performs well. This is not the case for all parts of $f$ but it holds for some 'small' parts of $f$ that appear in the splitting introduced in Lemma 3.1.

Next we deal with the remaining 'large' parts of $f$. We will first analyze the approximation for a single $\beta$. To this end fix $\beta \in \mathbb{R}^d$ and set $z = X\beta$. Our goal is to approximate $\|z^+\|_1 := \sum_{i:z_i>0} z_i$ where $z^+ \in \mathbb{R}^n_{\geq 0}$ is the vector that we get by setting all negative coordinates of $z$ to 0. We assume w.l.o.g. that $\|z\|_1 = 1$. We can do this since $v \mapsto \|v^+\|_1$ is absolutely homogeneous. In order to prove that $\|(Sz)^+\|_1$ approximates $\|z^+\|_1$ well, we define weight classes: given $q \in \mathbb{N}$ we set $W_q^+ = \{i \in [n] \mid z_i \in (2^{-q-1}, 2^{-q}]\}$. Our analysis applies with slight adaptations to $\ell_1$ regression preserving $\|z\|_1$ for the residual vector $z = X\beta - Y$. The analysis is entirely in the appendix due to the page limitations. We give a high level description for preserving $\|z^+\|_1$ needed for logistic loss.

**Contraction bounds** We set $q_m = \log_2(\frac{n(\mu+1)}{\varepsilon}) = O(\ln(n))$ since $n \geq \max\{\mu, \varepsilon^{-1}\}$. We say that $W_q^+$ is important if $\|W_q^+\|_1 \geq \varepsilon' := \frac{\varepsilon}{\mu q_m}$ and set $Q^* = \{q \leq q_m \mid W_q^+ \text{ is important }\}$. The idea is that the remaining weight classes can only have small contributions to $\|z^+\|_1$, so it suffices to analyze $Q^*$. To prove the contraction bound for $z$, i.e., that $\|(Sz)^+\|_1 \geq (1 - c\varepsilon)\|z^+\|_1$ holds for an absolute constant $c$, it suffices to show that the contributions of important weight classes are preserved. For a bucket $B$ we set $G(B) := \sum_{j \in B} z_j$ and $G^+(B) = \max\{G(B), 0\}$. In fact, we show that for each level $h$, there exists an 'inner' interval $Q_h = [q_h(2), q_h(3)]$ such that if $W_q^+$ for $q \in Q_h$ is important, then there exists a subset $W_q^* \subseteq W_q^+$ such that each element of $W_q^*$ is sampled at level $h$ and such that $\sum_{i \in W_q^*} G(B_i) \geq (1 - \varepsilon)\|W_q^+\|_1 \cdot p_h$, where $B_i$ is the bucket at level $h$ containing $z_i$. Since the weight of all buckets at level $h$ is equal to $p_h^{-1}$ we have that the contribution of $W_q^*$ is indeed at least $(1 - \varepsilon)\|W_q^+\|_1$. The choice of our parameters then guarantees that $\bigcup Q_h = \mathbb{N}$ and thus for any important weight class there is at least one level where it is well represented. Finally, we construct a net of size $|\mathcal{N}_k| = \exp(O(d \log(n)))$. We ensure that the contraction bound holds for each fixed net point $z \in \mathcal{N}_k$ with failure probability at most $\frac{\delta}{|N_k|}$ which will dominate – among other parameters – the size of our sketch. By a union bound, the contraction result holds for the entire net

with probability at least $1 - \delta$. The net is sufficiently fine, such that we can conclude the contraction bound by relating all other points $z = X\beta \in \mathbb{R}^n$ to their closest net point.

**Dilation bounds** We will also show that the expected contribution of any weight class is at most $2\|W_q^+\|_1$ or even less. To this end we increase the number of buckets $N$ and apply a random shift at level $0$, i.e., we choose the number of buckets at level $0$ randomly. We investigate again each level separately and prove that for each level $h$ there exists an 'outer' interval $Q_h' = [q_h(1), q_h(4)]$ such that for any $q \notin Q_h'$ the weight class $W_q$ makes no contribution at level $h$ at all. More specifically we show that no element of $W_q$ appears at level $h$ for $q < q_h(1)$ and that for any bucket $B$ at level $h$ that contains only elements of $\bigcup_{q>q_h(4)} W_q$ it holds that $G(B) \leq 0$.

Then we show that if $N$ is large enough it holds that for each $q \in \mathbb{N}$ there are at most two levels $h$ such that $q \in Q_h'$ and that the expected contribution of any weight class at any level is bounded by $\|W_q^+\|_1$. We conclude that the expected contribution of any weight class is at most $2\|W_q^+\|_1$. Increasing the size of $N$ increases the size of the 'inner' interval $[q_h(2), q_h(3)] =: Q_h \subset Q_h'$ while the size of $Q_h'$ remains (almost) unchanged such that $|Q_h'|/|Q_h|$ approaches $1$. As a consequence, this also decreases the number of indices $q \in \mathbb{N}$ that appear in two intervals of the form $Q_h'$. More precisely, we show that for each $k \in \mathbb{N}$ we can increase $N$ in such a way that only a $1/k$ fraction of the weight classes appear in two of those intervals. Note that all weight classes that appear only in a single $Q_h'$ have an expected contribution of $\|W_q^+\|_1$. Recall that *all* indices are considered on level $0$. This is handled by applying a random shift, implicitly setting $q_0(3)$ randomly in an appropriate way such that the expected contribution of *any* weight class $W_q^+$ is bounded by at most $(1 + 1/k)\|W_q^+\|_1$.

**Extension to variance-based regularized logistic regression** We show that our algorithm also approximates the variance well under the assumption that roughly $f_1(X\beta) \leq \ln(2)$. We stress that this assumption does not rule out the existence of good approximations. Indeed, even the minimizer is contained as observed in the preliminaries, since we have that $\min_{\beta \in \mathbb{R}^d} f(X\beta) \leq f(0) = f_1(0) = \ln(2)$. Focusing on a single $z = X\beta$, we need to show that $\sum_{i:z_i>0} z_i^2$ is approximated well, which is done very similarly to the analysis for $\sum_{i:z_i>0} z_i$ sketched above, but with several adaptions to account for the squared loss function. We note that the increased sketching dimension in terms of $\sqrt{n}$ comes from the inter norm inequality $\|x\|_1 \leq \sqrt{n}\|x\|_2$. Lemma E.14 in the appendix shows that this dependence can not be avoided using the `CountMin`-sketch. It does not rule out other methods that may allow a lower sketching dimension. We stress that other known standard sketches do not work for asymmetric functions since they confuse the signs of contributions leading to unbounded errors for our objective function or plain logistic regression, see (Munteanu et al., 2021).

## 4 EXPERIMENTS

We implemented our new sketching algorithm into the framework of Munteanu et al. (2021)[5]. Pseudocode can be found in Appendix G. The crucial difference is that at level $0$ of our sketch, each element gets mapped to multiple buckets instead of only one. Sketch (old) denotes the sketch used in (Munteanu et al., 2021) and is highlighted in red in the plots. Sketch$s$ is the sketch where each entry is mapped to $s \in \{2, 5, 10\}$ buckets at level $0$. Each sketch was run with $40$ repetitions for various target sizes. Real-world benchmark data was downloaded automatically by the Python scripts: the Covertype data consists of $581,012$ cartographic observations of different forests with $54$ features. The Webspam data consists of $350,000$ unigrams with $127$ features from web pages. The Kddcup data consists of $494,021$ network connections with $41$ features. On the real-world data we see in Figure 1 slightly improved performances over the previous sketch for Covertype and Webspam. On the Kddcup data we see a slightly weaker performance. We also see that increasing the sparsity parameter $s$ too much results in a worse performance, which is especially true for $\lambda > 0$ (partly in the appendix). This indicates that the variance term is large for Kddcup and the $\sqrt{n}$ dependence dominates the sketching size necessary to decrease the error in the squared variance-regularization term. We created a synthetic data set that has multiple heavy hitters for the sake of showing the benefits of the new sketch. A detailed description of our construction can be found in Appendix G, along with intuition why it is complicated for the old sketch, while our new sketch can handle it much better. The data set consists of $n = 40,000$ points and the dimension is $d = 100$. We see in Figure 1 (bottom left) that the increase in the number of buckets each elements is hashed to improves

---

[5]code available at `https://github.com/Tim907/oblivious_sketching_varreglogreg`

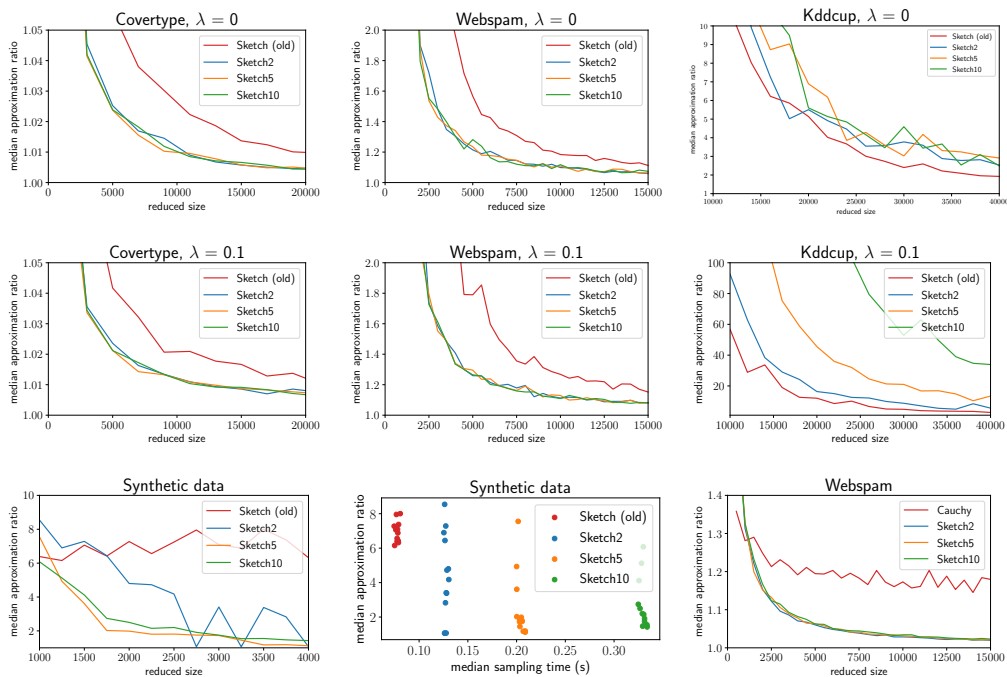

Figure 1: Comparison of median approximation ratios of the old sketch vs. the new sketch with various settings for the sparsity $s \in \{2, 5, 10\}$ as well as for the regularization parameter $\lambda \in \{0, .1\}$ for real-world benchmark data (rows 1-2). Comparison of median approximation ratios and sketching times (row 3, left, middle) for our synthetic data. Comparison to the Cauchy sketch (row 3, right).

from approximation ratios between 7 and 8 for the old sketch to 3 or even 2 approximations for the modified sketches. The sketching times are only slightly increased for larger values of $s$, allowing for fast processing time (bottom middle). We omitted the time plots for the other data sets and parameterizations since the general picture is consistently as expected: Sketch$s$ is almost $s$ times slower than Sketch (old). We added another comparison between our Sketch and the Cauchy sketch for an $\ell_1$ regression problem (bottom right). We see that the new sketch, using any degree of sparsity $s \in \{2, 5, 10\}$, outperforms the Cauchy sketch by a large margin in terms of approximation factor (while being a lot faster to apply than the dense matrix multiplication). More plots and discussion can be found in Appendix G. We also discuss stochastic gradient descent (SGD) together with supporting experiments in Appendix G. While SGD performs well on real-world data (though not better than sketching), it suffers from arbitrarily bad errors when applied to our synthetic data.

## 5    CONCLUSION

We obtain significantly improved bounds on the number of rows that are sufficient for obliviously sketching logistic regression on $\mu$-complex data up to an $O(1)$ factor. Our bounds are almost linear in terms of the dimension $d$ and a data dependent complexity parameter $\mu$ that bounds the complexity of data reduction techniques for logistic regression and related loss functions. Our results are achieved by modifying the sketching approach of Munteanu et al. (2021), which allows a change of perspective and facilitates a fine-grained analysis of the contributions of single levels in the sketch. As a result, we also develop the first oblivious sketch for obtaining a $(1 + \varepsilon)$-approximation, albeit with an exponential dependence on $1/\varepsilon$ which is likely to be required for our estimator due to corresponding hardness results on sketching $\ell_1$ norms. We also extend the analysis to work for a variance-based regularized version of logistic regression which combines the $\ell_1$ and $\ell_2$ related loss functions and is of great practical relevance for reducing generalization error in statistical learning. It remains a challenging open question whether we can further reduce the upper bounds to or below $O(\mu d)$ or increase the lower bounds from $\Omega(\mu + d)$ to or above $\Omega(\mu d)$. It would be interesting to study our sketching techniques under assumptions such as sparsity that allow to get below linear sketching dimension (Mai et al., 2023).

ACKNOWLEDGEMENTS

We thank the anonymous reviewers for their valuable comments. We thank Tim Novak for helping with the experiments. Alexander Munteanu & Simon Omlor were supported by the German Research Foundation (DFG), Collaborative Research Center SFB 876, project C4 and by the Dortmund Data Science Center (DoDSc). David P. Woodruff was partially supported by a Simons Investigator Award and by the Office of Naval Research (ONR) grant N00014-18-1-2562.

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

# A  OMITTED DETAILS FROM SECTION 2

For technical reasons we make the following assumption:

**Assumption A.1.** *We assume that:*

$$h_m = \min\left\{ i \in \mathbb{N} \mid \frac{M_i}{bN} \le 12\ln(n) \right\} \tag{1}$$

$$N \ge 32m_1^{1+c} q_m^{1+c} h_m^c \mu / \varepsilon^6 \tag{2}$$

$$b = \frac{N\varepsilon^5}{32m_1 q_m \mu} \ge \frac{18\mu}{\varepsilon} \tag{3}$$

$$m_1 = \ln(\delta^{-1}) + O(d\ln(n))) \tag{4}$$

$$p_u \ge \frac{64\mu m_1}{\varepsilon^2 n}. \tag{5}$$

*Here c is some constant used in the proof of Theorem 2. Since we want our sketch to have fewer than n rows we will also assume that $n \ge \varepsilon^{-1}, \mu, d, \delta^{-1}$. We also assume that $\varepsilon \le 1/4$.*

We will further use the following probability tools:

**Proposition A.1.** *[Bernstein's Inequality](Bernstein, 1924) Let $X_1, \ldots, X_n$ be independent zero-mean random variables. Suppose that $|X_i| \le M$ holds almost surely for all i. Then, for all positive t it holds that*

$$\mathbb{P}\left( \sum_{i=1}^n X_i \ge t \right) \le \exp\left( -\frac{\frac{1}{2}t^2}{\sum_{i=1}^n \mathbb{E}\left[X_i^2\right] + \frac{1}{3}Mt} \right).$$

**Proposition A.2** (Chernoff bound (Chernoff, 1952)). *Let $X = \sum_{i=1}^n X_i$, where $X_i = 1$ with probability $p_i$ and $X_i = 0$ with probability $1 - p_i$, and all $X_i$ are independent. Let $\mu = \mathbb{E}(X) = \sum_{i=1}^n p_i$. Then for all $\delta \in [0,1]$ it holds that*

$$P(|X - \mu| \ge \delta\mu) \le 2\exp(-\delta^2\mu/3)$$

*and for any $\delta > 1$ it holds that*

$$P(|X - \mu| \ge \delta\mu) \le 2\exp(-\delta\mu/3)$$

**Lemma A.3.** *Let y be a binomially distributed random variable with parameters $n, p$. Let $n' \in \mathbb{N}$. Then if $n' \ge pn$ we have that*

$$P(|y - pn| > n') \le 2\exp\left(-n'/3\right)$$

*Else if $n' = \varepsilon pn$ we have that*

$$P(|y - pn| > n') \le 2\exp\left(-\varepsilon n'/3\right)$$

*Proof of Lemma A.3.* Note that $\mathbb{E}(y) = pn$. Using the Chernoff bound we get that if $n' \ge pn$

$$P(|y - pn| > n') \le 2\exp(-(n'/np)np/3) = 2\exp(-n'/3).$$

If $n' = \varepsilon pn$ the Chernoff bound implies that

$$P(|y - pn| > n') \le 2\exp(-\varepsilon^2 np/3) = 2\exp\left(-\varepsilon n'/3\right).$$

$\square$

Lemma 3.1 in the main body splits the objective into 'large' and 'small' parts which we handle separately.

*Proof of Lemma 3.1.* Note that for $r \in \mathbb{R}$ it holds that

$$\ell(r) = \ln\left(1 + e^r\right) = \ln\left(\left(e^{-r} + 1\right)e^r\right)$$
$$= \ln\left(e^{-r} + 1\right) + \ln(e^r) = \ell(-r) + r.$$

Now the first equation follows immediately by $\ell(x_i\beta) = x_i\beta + \ell(-x_i\beta) = |x_i\beta| + \ell(-|x_i\beta|)$ for $x_i\beta > 0$ and $\ell(x_i\beta) = \ell(-|x_i\beta|)$ for $x_i\beta \le 0$. Further we have that $(x_i\beta + \ell(-x_i\beta))^2 = (x_i\beta)^2 + 2\ell(-x_i\beta)x_i\beta + \ell(-x_i\beta)^2$ Thus the second equality follows by substituting $\ell(x_i\beta)^2$ with $|x_i\beta|^2 + 2\ell(-|x_i\beta|)|x_i\beta| + \ell(-|x_i\beta|)^2 = (x_i\beta)^2 + 2\ell(-x_i\beta)x_i\beta + \ell(-x_i\beta)^2$ for $x_i\beta > 0$ and $\ell(-|x_i\beta|)^2$ for $x_i\beta \le 0$. $\square$

## B    ESTIMATING THE SMALL PARTS OF $f$

We can bound the 'small' parts using the following lemma:

**Lemma B.1.** *For arbitrary $i \in [n]$ it holds that $\ell(-|x_i\beta|) < 1$ and also $2\ell(-|x_i\beta|)|x_i\beta| + \ell(-|x_i\beta|)^2 \leq 3$.*

*Proof.* First observe that $\ell(-|x_i\beta|) \leq \ell(0) = \ln(2) < 1$, proving the first part of the lemma.

Next note that

$$\ell(-|x_i\beta|) = \ln(1 + \exp(-|x_i\beta|)) = \int_1^{1+\exp(-|x_i\beta|)} \frac{1}{t}\, dt$$
$$\leq \int_1^{1+\exp(-|x_i\beta|)} 1\, dt = \exp(-|x_i\beta|).$$

Using that $\ln(t) \leq |t|$ for all $t > 0$ we conclude that

$$\ell(-|x_i\beta|)|x_i\beta| \leq \exp(\ln(|x_i\beta|) - |x_i\beta|) \leq e^0 = 1.$$

Now combining everything we get that

$$2\ell(-|x_i\beta|)|x_i\beta| + \ell(-|x_i\beta|)^2 \leq 2 + 1^2 \leq 3.$$

$\square$

Next we note that the optimal value of $f(X\beta)$ is bounded from below:

**Lemma B.2.** *(Munteanu et al., 2021) For all $\beta \in \mathbb{R}^d$ it holds that $nf(X\beta) \geq nf_1(X\beta) \geq \frac{n}{2\mu}(1 + \ln(\mu)) = \Omega\left(\frac{n}{\mu}(1 + \ln(\mu))\right)$.*

We use the previous two lemmas to show that our sketch approximates the given parts of $f$ well enough with high probability. To this end, we set $g_1(t) = \ell(-|t|)$, $g_2(t) = 2\ell(-|t|)|t| + \ell(-|2t|)$ and $g(t) = g_1(t) + \lambda g_2(t)$.

**Lemma B.3.** *Given any $\beta \in \mathbb{R}^d$ with failure probability at most $2\exp(-m_1)$ the event $\mathcal{E}_0$ holds that*

$$\left| \sum_{i=1}^{n'} w_i g(x_i'\beta) - \sum_{i=1}^{n} g(x_i\beta) \right| \leq \varepsilon \cdot \max\left\{ \sum_{i=1}^{n} g(x_i\beta), \frac{n}{2\mu} \right\} \leq \varepsilon f(X\beta).$$

*Proof of Lemma B.3.* The total weight of all buckets in a level less than $h_m$ is at most $\sum_{h=1}^{h_{\max}} b^{-h} = b^{-1} \cdot \frac{1 - b^{-h_{\max}}}{1 - b^{-1}} \leq \frac{2}{b} \leq \frac{\varepsilon}{6\mu}$. Now let $k \in \{1, 2\}$. For $i \in [n]$, consider the random variable $X_i = g_k(z_i)$ if $z_i$ is at level $h_m$, and $X_i = 0$ otherwise. Then we have

$$E = \mathbb{E}\left( \sum_{i=1}^{n} X_i \right) = \sum_{i=1}^{n} p_u g_k(z_i) = p_u \sum_{i=1}^{n} g_k(x_i\beta).$$

Further we have $X_i \leq 3$ by Lemma B.1. It holds that

$$\mathbb{E}\left( \sum_{i=1}^{n} X_i^2 \right) = \sum_{i=1}^{n} p_u g_k(z_i)^2 \leq p_u \sum_{i=1}^{n} 3g(x_i\beta) = 3E.$$

We set

$$L = p_u \cdot \max\left\{ \sum_{i=1}^{n} g_k(x_i\beta), \frac{n}{2\mu} \right\} \geq E.$$

By Assumption A.1 we have that $p_u \geq \frac{64\mu m_1}{\varepsilon^2 n}$. Thus, using Bernstein's inequality we get that

$$P\left( \left| \sum_{i=1}^{n} X_i - E \right| \geq \frac{\varepsilon}{2} \cdot L \right) \leq \exp\left( \frac{-\varepsilon^2 L^2/8}{3E + E} \right)$$

$$= \exp\left(\frac{-\varepsilon^2 L}{32}\right)$$

$$\leq \exp\left(\frac{-\varepsilon^2 p_u n/\mu}{64}\right)$$

$$\leq \exp(-m_1).$$

Using the union bound for $k = 1$ and $k = 2$ yields that

$$P\left(\left|\sum_{i=1}^{n'} w_i g(x_i'\beta) - \sum_{i=1}^{n} g(x_i\beta)\right| > \varepsilon \cdot \max\left\{\sum_{i=1}^{n} g(x_i\beta), \frac{n}{2\mu}\right\}\right) \leq 2\exp(-m_1).$$

By Lemma B.2 we have $f(X\beta) \geq \frac{n}{2\mu}$. It also holds that $f(X\beta) \geq \sum_{i=1}^{n} g(x_i\beta)$. We thus conclude that $\varepsilon \cdot \max\left\{\sum_{i=1}^{n} g(x_i\beta), \frac{n}{2\mu}\right\} \leq \varepsilon f(X\beta)$. $\qquad\square$

## C  ESTIMATING THE LARGE PARTS $\|z\|_1$ AND $\|z^+\|_1$

**Lemma C.1.** *It holds that* $\sum_{q \in Q^*} \|W_q^+\|_1 \geq (1 - 2\varepsilon)\|z^+\|_1$.

*Proof of Lemma C.1.* First note that

$$\sum_{z_i < 2^{-q_m}} z_i \leq n \cdot \frac{\varepsilon}{(\mu+1)n} = \varepsilon/(\mu+1).$$

Second note that $\sum_{q \leq q_m, q \notin Q^*} \|W_q^+\|_1 \leq q_m \cdot \frac{\varepsilon}{(\mu+1)q_m} \leq \varepsilon/(\mu+1)$. By the $\mu$-condition we have that $\|z^-\|_1 \leq \mu\|z^+\|_1$ and thus we get that $1 = \|z^-\|_1 + \|z^+\|_1 \leq \mu\|z^+\|_1 + \|z^+\|_1$. Consequently, $\|z^+\|_1 \geq \frac{1}{\mu+1}$ and $\sum_{q \in Q^*} \|W_q^+\|_1 \geq \|z^+\|_1 - \frac{2\varepsilon}{(\mu+1)} \geq (1 - 2\varepsilon)\|z^+\|_1$. $\qquad\square$

### C.1  ANALYSIS FOR A SINGLE LEVEL

Fix $h \in [0, h_m]$. First consider the number of elements at a fixed level $h$. We can view it as a binomial random variable with parameters $n$ and $p_h$ since the probability for any row to appear at level $h$ is $p_h$. Since we fix $h$ in this subsection, we set $M = M_h = p_h n$, $p = p_h = \frac{M}{n}$ and $N = N_h$. We set $U \subset [n]$ to be the set of elements that are sampled at level $h$. We also set $\mu_z = \frac{\sum_{z_i \leq 0} |z_i|}{\sum_{z_i > 0} |z_i|} \leq \mu$.

This and the following subsection are dedicated to proving the existence of bounds $q_h(1)$, $q_h(2)$, $q_h(3)$ and $q_h(4)$ as described in the high level overview, Section 3. More precisely we show the following:

**Lemma C.2.** *With probability at least* $1 - \frac{\delta}{h_m}$ *the weight classes* $W_q$ *for* $q \geq q_{(M,N)}(4) := \log_2(\gamma_2^{-1}) := \log_2(\frac{2N\ln(Nh_{\max}/\delta)}{p\varepsilon^2})$ *and* $q \leq q_{(M,N)}(1) := \log_2(\frac{\mu_z \delta}{ph_{\max}})$ *have zero contribution to* $\sum_B G^+(B)$, *i.e., for any bucket* $B$ *we have* $\sum_{z_i \in B \setminus I_r} z_i \leq 0$ *where* $I_r = \{i \in [n] \mid z_i \in W_q, q \in [q_{(M,N)}(1), q_{(M,N)}(4)]\}$. *Further, with failure probability at most* $\exp(-\Omega(m_1))$ *there exists, for each* $\log_2(\frac{8q_m \mu_z m_1}{\varepsilon^3 p})) =: q_{(M,N)}(2) \leq q \leq q_{(M,N)}(3) := \log_2(\frac{N\varepsilon^2}{4p})$, *a set* $W_q^*$ *such that* $\sum_{i \in W_q^*} G(B_i) \geq (1-\varepsilon)^2 \|W_q^+\|_1 \cdot \frac{M}{n}$. *It thus holds that*

$$q_{(M,N)}(2) - q_{(M,N)}(1) = \log_2\left(\frac{8q_m m_1 h_m}{\varepsilon^3 \delta}\right)$$

$$q_{(M,N)}(3) - q_{(M,N)}(2) = \log_2\left(\frac{N\varepsilon^5}{32m_1\mu q_m}\right) =: \log_2(b)$$

$$q_{(M,N)}(4) - q_{(M,N)}(3) = \log_2\left(\frac{8\ln(Nh_m/\delta)}{\varepsilon^4}\right).$$

If $N = M$ then we set $q_{(M,N)}(3) = q_{(M,N)}(4) = \infty$. If $M = n$ then we set $q_{(M,N)}(1) = q_{(M,N)}(2) = 0$. We set $q_h(i) = q_{(M_h,N_h)}(i)$ for $i \in \{1,2,3,4\}$ and $Q_h = [q_h(2), q_h(3)]$ to be the

well-approximated weight classes, and $R_h = [q_h(1), q_h(4)]$ to be the relevant weight classes at level $h$.

We further define the following threshold and set:

$$\gamma_1 := \frac{p}{3m_1}$$
$$Y_1 := \{i \in [n] \mid |z_i| \geq \gamma_1\}$$

Here $Y_1$ is the 'set of large elements'. We set $\mathcal{B}_h$ to be the set of all buckets at level $h$. Recall that $m_1 \in \mathbb{R}$ is a lower bound on the negative logarithm of the failure probability, which we will need later when union bounding over all failure probabilities. Also recall that $G(B) = \sum_{i \in B} z_i$ is the sum of all rows in a bucket $B$. The following lemma yields the inner bounds, i.e., bounds for $q_h(2)$ and $q_h(3)$, which are the weight class indices that are well represented by $U$. The first two items show that there are at most $\varepsilon N$ buckets at level $h$ that either contain a large element or have a large sum of small contributions. The third item shows that if $W_q$ has sufficiently many elements, then there exists a large subset $W_q^*$ where each element is in a bucket with no other large entry such that $\|W_q^*\|_1$ is close to $\|W_q^+\|_1 \cdot \frac{M}{n}$. The fourth item shows that $\sum_{z_i \in W_q^*} G(B_i)$ is close to $\|W_q^*\|_1$.

**Lemma C.3.** *The following hold:*

1) *$|Y_1 \cap U| \leq \varepsilon N/2$ with failure probability at most $\exp(-m_1)$;*

2) *Let $\mathcal{B} = \{B \in \mathcal{B}_h \mid \sum_{i \in B \setminus Y_1} |z_i| \leq \frac{4p}{\varepsilon N}\}$. Then $|\mathcal{B}| \geq (1 - \frac{\varepsilon}{2})N$ with failure probability at most $\exp(-m_1)$;*

3) *Assume that $q \geq \log_2(\frac{8 q_m \mu_z m_1}{\varepsilon^3 p})$ and that $W_q^+$ is important or $|W_q| \geq 8 m_1 \varepsilon^{-2} \cdot p^{-1}$. Then with failure probability at most $\exp(-m_1)$ there exists $W_q^* \subset W_q^+ \cap \mathcal{B}$ such that $\|W_q^*\|_1 \geq (1 - \varepsilon)^2 \|W_q^+\|_1 \cdot p$ and each element of $W_q^*$ is in a bucket in $\mathcal{B}$ containing no other element of $Y_1$;*

4) *If $q \leq \log_2(\frac{N \varepsilon^2}{4p})$ and $W_q^*$ as in 3) exists, then with failure probability at most $\exp(-m_1)$ it holds that $\sum_{i \in W_q^*} G(B_i) \geq (1 - \varepsilon) \|W_q^*\|_1$.*

*Proof.* 1) Note that $|Y_1| \leq \gamma_1^{-1}$ since $\|z\|_1 = 1$ and that we can view $|Y_1 \cap U|$ as a binomial random variable with parameters $|Y_1|$ and $p = \frac{M}{n}$. Thus, the expected number of elements of $Y_1$ at level $h$ is bounded by $|Y_1| \cdot \frac{M}{n} \leq \frac{p}{\gamma_1} = 3m_1 \leq \frac{\varepsilon N}{4}$ since $N \geq 12 m_1$ (see Assumption A.1). Thus, we get by Lemma A.3 that

$$P\left(|Y_1 \cap U| \geq \frac{\varepsilon N}{2}\right) \leq P\left(|Y_1 \cap U| - |Y_1| \cdot p \geq \frac{\varepsilon N}{4}\right) \leq P\left(|Y_1 \cap U| - |Y_1| \cdot p \geq 3m_1\right)$$
$$\leq \exp\left(-3m_1/3\right) \leq \exp(-m_1).$$

2) For $i \in T = [n] \setminus Y_1$ we set $X_i = |z_i|$ if $i \in U$ and $X_i = 0$ otherwise. Since $\sum_{i \in T} |z_i| \leq \|z\|_1 = 1$ we have that $\mathbb{E}(\sum_{i \in T} X_i) = p \cdot \sum_{i \in T} |z_i| \leq p$. Since all 'large elements' are in $Y_1$ we have that $X_i < \gamma_1$ for all $i \in [n]$ and thus

$$\mathbb{E}\left(\sum_{i \in T} X_i^2\right) = \sum_{i \in T} p|z_i|^2 \leq \sum_{i \in T} p\gamma_1|z_i| = p\gamma_1 \sum_{i \in T} |z_i| \leq p\gamma_1.$$

Using Bernstein's inequality we get

$$P\left(\sum_{i \in T} X_i \geq 2p\right) \leq \exp\left(-\frac{p^2/2}{p\gamma_1 + p\gamma_1/3}\right) \leq \exp\left(-\frac{p}{3\gamma_1}\right) = \exp(-m_1).$$

This implies that $\sum_{i \in T} X_i \leq 2p$ with failure probability at most $\exp(-m_1)$. Now if $\sum_{i \in T} X_i \leq 2p$ then there can be at most $\frac{\varepsilon N}{2}$ buckets $B$ with $G(B \setminus Y_1) \geq \frac{4p}{\varepsilon N}$.

3) First note that if $q \geq \log_2(\frac{8q_m\mu_z m_1}{\varepsilon^3 p})$ is important then $2^{-q} \cdot |W_q^+| \geq \|W_q^+\|_1 \geq \frac{\varepsilon}{q_m\mu_z}$, which implies that $|W_q^+| \geq \frac{2^q\varepsilon}{q_m\mu_z} \geq 8m_1\varepsilon^{-2} \cdot p^{-1}$. Assume that all entries of $Y_1 \setminus W_q^+$ have been assigned and let $\mathcal{B}' \subset \mathcal{B}$ be the buckets of $\mathcal{B}$ with no elements from $Y_1 \setminus W_q^+$. By 1) and 2) there are at least $(1-\varepsilon)N$ buckets in $\mathcal{B}'$. For $z_i \in W_q^+$ consider the random variable that takes the value $Z_i = z_i$ if $i \in \bigcup_{B \in \mathcal{B}'} B$ and $Z_i = 0$ otherwise. Set $Z = \sum_{z_i \in W_q^+} Z_i$. We have $Z_i = z_i$ if element $i$ is sampled at level $h$ and sent to a bucket in $\mathcal{B}'$, which happens with probability at least $p \cdot \frac{(1-\varepsilon)N}{N} = (1-\varepsilon)p$. We thus have for the expected value of $Z$ that

$$\mathbb{E}(Z) \geq (1-\varepsilon)p \cdot \|W_q^+\|_1 \geq (1-\varepsilon)p \cdot 2^{-q-1} \cdot |W_q^+| \geq (1-\varepsilon) \cdot 2^{-q-1} \cdot 8m_1\varepsilon^{-2}$$
$$\geq 2^{-q} \cdot 3m_1\varepsilon^{-2}.$$

Further, the maximum value of any $Z_i$ is $2^{-q}$ and the probability that $Z_i = z_i$ is upper bounded by $p$. Consequently, the variance of $Z$ is bounded by

$$\sum_{z_i \in W_q^+} \mathbb{E}(Z_i^2) \leq \sum_{z_i \in W_q^+} pz_i^2 \leq 2^{-q} \sum_{z_i \in W_q^+} pz_i = 2^{-q}\mathbb{E}(Z).$$

Using Bernstein's inequality we get that

$$P\left(Z < (1-\varepsilon)^2 p \cdot \|W_q^+\|_1\right) \leq P\left(Z - \mathbb{E}(Z) > \varepsilon\mathbb{E}(Z)\right)$$
$$\leq \exp\left(\frac{-\varepsilon^2\mathbb{E}(Z)^2/2}{2^{-q}\mathbb{E}(Z) + 2^{-q}\varepsilon\mathbb{E}(Z)/3}\right)$$
$$\leq \exp\left(\frac{-\varepsilon^2\mathbb{E}(Z)}{3 \cdot 2^{-q}}\right)$$
$$\leq \exp\left(-m_1\right).$$

We set $W_q^* = \{z_i \in W_q^+ \mid Z_i = z_i\}$.

4) By 2) and 3) we have that any entry $z_i \in W_q^*$ is in a bucket $B$ with $\sum_{j \in B \setminus \{i\}} |z_j| \leq \frac{4p}{\varepsilon N}$. Thus, we have for $z_i \geq \frac{4p}{\varepsilon^2 N}$ that $\sum_{j \in B_i} z_j \geq z_i - \frac{4p}{\varepsilon N} \geq (1-\varepsilon)z_i$. Now we conclude

$$\sum_{i \in W_q^*} G(B_i) \geq \sum_{i \in W_q^*} (1-\varepsilon)z_i = (1-\varepsilon)\|W_q^*\|_1.$$

$\square$

Note that if all buckets contain only a single element then we can remove the condition $q \leq \log_2(\frac{N\varepsilon^2}{4p})$. Hence, we can set $q_{(M,N)}(3) = q_{(M,N)}(4) = \infty$ if $N = M$ (respectively, $h = h_m$).

For the outer bounds, i.e., the borders of the interval of weight classes that can have a non-negligible contribution to $U$, we need the following parameters defining the set of small elements:

$$\gamma_2 := \frac{p\varepsilon^2}{3N\ln(Nh_{\max}/\delta)}$$
$$Y_2 = \{i \in [n] \mid |z_i| \leq \gamma_2\}$$

We further set $E$ to be the expected value of an entry chosen uniformly at random from $Y_2$.

**Lemma C.4.** *The following hold:*

1) *If $E \leq -\varepsilon/n$, then for any bucket $B$ that contains only elements of $Y_2$, we have $G(B) = \sum_{i \in B} z_i \leq 0$ with failure probability at most $\frac{\delta}{Nh_{\max}}$.*

2) *$U$ contains no element $i$ with $z_i \geq \frac{ph_{\max}}{\delta}$ with failure probability at most $\frac{\mu_z\delta}{h_{\max}}$.*

*Proof.* 1) First consider a single bucket $B$ containing only elements of $Y_2$. For $i \in [n]$, let $X_i$ be a random variable that attains the value $X_i = z_i$ if $i \in B$ and $X_i = 0$ otherwise. The expected value

of $G(B) = \sum_{i \in [n]} X_i$ is $E' := n \cdot \frac{p}{N} \cdot E \leq -\frac{p\varepsilon}{N}$. Further, we have that

$$\mathbb{E}\left(\sum_{i \in [n]} X_i^2\right) = \sum_{i \in Y_2} \frac{p}{N} \cdot z_i^2 \leq \gamma_2 \cdot \sum_{i \in Y_2} \frac{p}{N} \cdot |z_i| = \gamma_2 \frac{p}{N}$$

since all $X_i$ are bounded by $\gamma_2$ by assumption. Thus, applying Bernstein's inequality yields

$$P(G(B) > 0) \leq P\left(\sum_{i \in [n]} X_i - E' \geq |E'|\right) \leq \exp\left(\frac{-|E'|^2/2}{\gamma_2 \frac{p}{N} + \gamma_2 |E'|/3}\right)$$

$$\leq \exp\left(\frac{-\varepsilon \cdot p/(N)}{2\gamma_2(p/(N|E'|) + 1/3)}\right)$$

$$\leq \exp\left(\frac{-\varepsilon \cdot p/(N)}{2\gamma_2(\varepsilon^{-1} + 1/3)}\right)$$

$$\leq \exp\left(\frac{-\varepsilon^2 \cdot p/(N)}{3\gamma_2}\right)$$

$$\leq \exp\left(-\ln\left(\frac{Nh_{\max}}{\delta}\right)\right)$$

$$= \frac{\delta}{Nh_{\max}}.$$

2) Recall that $\sum_{z_i > 0} z_i \leq 1/\mu_z$. Thus, there are at most $\frac{n\delta}{\mu_z M h_{\max}}$ entries with $z_i \geq \frac{M h_{\max}}{n\delta}$. The expected number of those entries in $U$ is thus at most $\frac{n\delta}{\mu_z M h_{\max}} \cdot \frac{M}{n} \leq \frac{\delta}{h_{\max}}$, which also upper bounds the probability of at least one entry with $z_i \geq \frac{M h_{\max}}{n\delta}$ being contained in $U$. $\square$

Putting both lemmas together we get all bounds $q_h(i)$ except $q_0(2)$, which will be handled in the next subsection.

## C.2 Heavy hitters

In this subsection we will analyze the level containing all entries. Our goal is to show that we can indeed set $q_0(2) = 0$ in Lemma C.2. Let $U$ be as before and assume that $M = n$. Let $Q_H = \{q \in Q_0 \mid |W_q| \geq 8m_1\varepsilon^{-2}\}$ where $Q_0 = \{q \leq \log_2(\frac{8q_m\mu m_1}{\varepsilon^3})\}$. We set $H = \bigcup_{q \in Q_H} W_q$ to be the class of heavy hitters.

We let $u \in \mathbb{R}_{\geq 0}^n$ denote the vector whose coordinates $u_i$ denote the $i$-th $\ell_1$-leverage scores, i.e., $u_i = \max_{\beta \in \mathbb{R}^d} \frac{|x_i\beta|}{\sum_{j \in [n]} |x_j\beta|}$.

**Lemma C.5.** *(Munteanu et al., 2021) If $u_i$ is the $k$-th largest coordinate of $u$, then for $z$ in the subspace spanned by the columns of $X$ it holds that $|z_i| \leq \frac{d}{k}\|z\|_1$. Further, it holds that $\sum_{i=1}^n u_i \leq d$.*

**Lemma C.6.** *Let $Y_3 = \{i \mid u_i \geq \gamma_3\}$ and $N_1 = |Y_3|$ where $\gamma_3 = \frac{\varepsilon^3}{8q_m\mu m_1}$. Further, for $j \in Y_3$ let $\mathcal{C}_j = \{B \mid \sum_{i \in B\setminus\{j\}} u_i \geq \varepsilon\gamma_3\}$. Then for all $j \in Y_3$ we have that $|\mathcal{C}_j|$ is bounded by $N_2 = d(\varepsilon\gamma_3)^{-1}$. Further if $N \geq N_1 N_2 \kappa^{-1}$ for $\kappa \in (0, 1/2)$, then with probability $1 - 2\kappa$, each member of $Y_3$ is in a bucket in $\mathcal{B}_0 \setminus \mathcal{C}_j$.*

*Proof.* Since by Lemma C.5 it holds that $\sum_{i=1}^n u_i \leq d$, there can be at most $d\frac{1}{\varepsilon\gamma_3} = N_2$ buckets $B$ with $\sum_{i \in B} u_i \geq \varepsilon\gamma_3$. In particular this implies that $|\mathcal{C}_j| \leq N_2$. The probability of any element of $j \in Y_3$ getting assigned to a bucket in $\mathcal{C}_j$ is at most $\frac{N_2}{N} \leq \frac{\kappa}{N_1}$. Using the union bound the probability that any element $j \in Y_3$ is assigned to a bucket in $\mathcal{B}_0 \setminus \mathcal{C}_j$ is at most $\kappa$. $\square$

We apply Lemma C.6 with $\kappa = \delta$. We denote by $\mathcal{E}_1$ the event that all coordinates in $j \in Y_3$ are in a bucket in $\mathcal{B}_0 \setminus \mathcal{C}_j$. By Lemma C.6 $\mathcal{E}_1$ holds with probability at least $1 - \delta$ for an appropriate

$N = N_0' = N_1 N_2 \delta^{-1} = \frac{64 d^2 q_m^2 \mu^2 m_1^2}{\delta \varepsilon^7} = \mathcal{O}(\frac{d^2 q_m^2 \mu^2 m_1^2}{\delta \varepsilon^7})$. For any entry $z_i \in H$ we have $z_i \geq \gamma_3$ and thus by Lemma C.5, we have $i \in Y_3$. It remains to show that the remaining entries in the buckets containing a heavy hitter only have a small contribution.

**Lemma C.7.** *Assume $\mathcal{E}_1$ holds. Then for any $z_i \in H$ we have $G(B_i) \geq (1-\varepsilon)z_i$.*

*Proof.* Let $z_i \in H$. Note that by Lemma C.5 $i \in Y_3$. By $\mathcal{E}_1$ we have that $\sum_{j \in B \setminus \{i\}} u_j \leq \varepsilon \gamma_3 \leq \varepsilon z_i$. We conclude that

$$G(B_i) \geq z_i - \sum_{j \in B_i \setminus \{i\}} |z_j| \geq z_i - \sum_{j \in B_i \setminus \{i\}} u_j \geq z_i - \varepsilon z_i \geq (1-\varepsilon)z_i.$$

$\square$

## C.3 CONTRACTION BOUNDS FOR A SINGLE POINT

We set $U_h$ to be the rows $z_i$ sampled at level $h$. Combining previous subsections we get the following lemma:

**Lemma C.8.** *Assume that $\mathcal{E}_1$ holds. Denote by $z_i'$ the $i$-th row of $SX\beta$ for $i \in n'$. Then with failure probability at most $(2h_m + 2q_m)e^{-m_1}$ it holds that*

$$\sum_{i \in n', z_i' \geq 0} w_i z_i' \geq (1 - 4\varepsilon)\|(X\beta)^+\|_1.$$

*Proof.* By Lemma C.3 and Lemma C.7 we have that for each important weight class $W_q^+$ there exists a subset $W_q^* \subseteq U_h$ with $\sum_{i \in W_q^*} G(B_i) \geq (1-\varepsilon)^2 \|W_q^+\|_1 p_h$. For $q \in Q_H$ we can set $W_q^* = W_q^+$. Then using Lemma C.1 we get

$$\sum_{i \in n', z_i' \geq 0} w_i z_i' \geq \sum_{q \in Q^*} p_h^{-1} \sum_{i \in W_q^*} G(B_i)$$

$$\geq \sum_{q \in Q^*} (1-\varepsilon)^2 \|W_q^+\|_1$$

$$\geq (1 - 2\varepsilon)(1-\varepsilon)^2 \|(X\beta)^+\|_1 \geq (1 - 4\varepsilon)\|(X\beta)^+\|_1.$$

$\square$

## C.4 DILATION BOUNDS

Given $\beta \in \mathbb{R}^d$ and $z = X\beta$ set $Z_0 = Z_0(\beta) \subset Z = \{z_1, \ldots, z_n\}$ to be the set of the $(1-\varepsilon)n$ largest entries ordered by absolute value. In other words, we remove the $\varepsilon n$ smallest entries. Similarly we set $Z_1 = Z_1(\beta) \subset Z$ to be the set of the $(1-2\varepsilon)n$ largest entries. Again we assume that $\|z\|_1 = 1$. Our next goal is to show that if $f(z)$ is small then $\sum_{z_i \in Z_0} z_i$ remains negative even if we remove the smallest entries. Here small means negative with large absolute value. This shows that the assumption of Lemma C.4 1) is fulfilled.

**Lemma C.9.** *If $f(X\beta) < (1 - 2\varepsilon)f(0)$ then it holds that*

$$\sum_{z_i \in Z_0, z_i \leq 0} |z_i| \geq (1 + \varepsilon) \sum_{z_i \geq 0} |z_i|$$

*Proof.* Let $X_1$ denote the matrix $X$ where the columns not corresponding to an entry of $Z_1$ are removed. We denote by $\tilde{f}$ the function $f$ restricted to $|Z_1|$ entries, i.e., $\tilde{f}(X\beta) = \sum_{x_i \in X_1} \ell(x_i\beta)$. Since $\ell$ is always larger than 0, removing $2\varepsilon n$ entries can only reduce $f$. We thus have that

$$\tilde{f}(0) = (1 - 2\varepsilon)f(0) \geq f(X\beta) = f(Z) \geq \tilde{f}(Z_1).$$

Now consider the function $\phi(r) = \tilde{f}(r \cdot X\beta)$. Note that the derivative of $\phi$ at zero is given by $\phi'(0) = \sum_{x_i \in X_1} \frac{e^0}{e^0+1} \cdot x_i\beta = \frac{1}{2} \cdot \sum_{z_i \in Z_1} z_i$. Since $\tilde{f}$ is convex $\phi$ is also convex. In particular this

means that $\tilde{f}(X\beta) < \tilde{f}(0)$ implies $\phi'(0) < 0$. Thus it must hold that $\sum_{z_i \in Z_1} z_i < 0$, or equivalently, $\sum_{z_i \in Z_1, z_i < 0} |z_i| > \sum_{z_i \in Z_1, z_i > 0} |z_i|$. Since all entries in $Z_0 \setminus Z_1$ are less than or equal to any entry in $Z_0$, we have that

$$\sum_{z_i \in Z_0, z_i < 0} |z_i| \geq \frac{1}{1-\varepsilon} \sum_{z_i \in Z_1, z_i < 0} |z_i| \geq (1+\varepsilon) \sum_{z_i > 0} |z_i|.$$

$\square$

The following lemma gives us an upper bound on the expected value of $G^+(Z)$.

**Lemma C.10.** *If for all $i \leq h_m - 1$ it holds that $q_{(M_i N_i)}(4) < q_{(M_{i+k} N_{i+k})}(1)$ and $N_0 \geq N_0'$, then the expected contribution of any weight class $W_q^+$ is at most $k \cdot \|W_q^+\|_1$.*

*Proof.* Consider any weight class $W_q^+$. For any level $h$ it follows by Lemma C.2 that if $q \notin [q_{(M_h N_h)}(1), q_{(M_h N_h)}(4)]$ then $W_q^+$ has zero contribution at level $h$, i.e., either there are no elements of $W_q^+$ at level $h$ or we have $W_q^+ \subset Y_1$ and for any bucket $B$ of level $h$ it holds that $\sum_{i \in Y_1 \cap B} z_i \leq 0$. At any level the expected contribution of $W_q^+$ is bounded by $p_h^{-1} \cdot \sum_{i \in W_q^+} p_h z_i = \|W_q^+\|_1$. This upper bound would be tight if all entries of $Z$ were positive. Hence, the expected contribution of $W_q^+$ is upper bounded by the number of levels $h$ with $q \in [q_{(M_h N_h)}(1), q_{(M_h N_h)}(4)]$. Since $q_{(M_h N_h)}(1)$ and $q_{(M_h N_h)}(4)$ are monotonically increasing in $h$, it follows that if $q_{(M_i N_i)}(4) < q_{(M_{i+k} N_{i+k})}(1)$ then any $q$ can be contained in at most $k$ intervals of the form $[q_{(M_h N_h)}(1), q_{(M_h N_h)}(4)]$, concluding the lemma. See Figure 2 for an illustration. $\square$

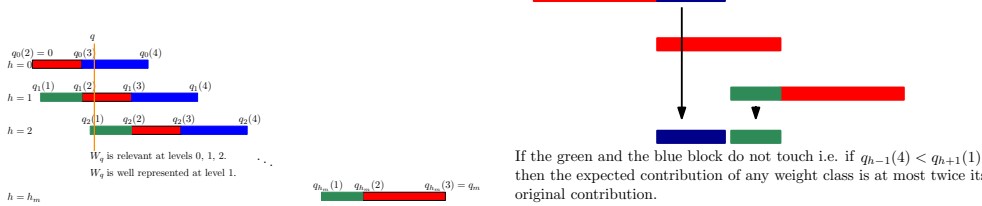

Figure 2: Illustration of Lemma C.10 and Lemma C.11.

Lemma C.10 can be used to show that the expected contribution of any weight class to $G^+(Z)$ is at most twice its total weight:

**Lemma C.11.** *If we choose $N_i = N := \max\{N_0', \frac{2048 m_1^2 \mu \ln(Nh_m/\delta) q_m^2 h_m}{\varepsilon^{12}\delta}\}$ for all $i \in [h_m]$, and $M_i$ solving the equation $q_{(M_{i-1}, N)}(3) = q_{(M_i, N)}(2)$ then the expected contribution of any weight class $W_q^+$ is at most $2\|W_q^+\|_1$.*

*Proof.* We set $q_i(j) = q_{(M_i N_i)}(j)$. We first show that $q_{(i+2)}(1) - q_i(4)$ can be expressed using the terms $q_{i+1}(3) - q_{i+1}(2)$, $(q_{i+2}(2) - q_{i+2}(1))$ and $(q_i(4) - q_i(3))$, which are the same for each $i$ if the number of buckets at each level is identical, i.e., for all $j \leq h_q$ it holds that $N_j = N_i$. Observe that

$$q_{(i+2)}(1) - q_i(4) = q_{i+2}(2) + q_{i+2}(1) - q_{i+2}(2) - (q_i(3) + q_i(4) - q_i(3))$$
$$= q_{i+2}(2) - q_i(3) - (q_{i+2}(2) - q_{i+2}(1)) - (q_i(4) - q_i(3))$$
$$= q_{i+1}(3) - q_{i+1}(2) - (q_{i+2}(2) - q_{i+2}(1)) - (q_i(4) - q_i(3)).$$

Figure 2 illustrates those three terms. Using Lemma C.2 we can bound the sum of the two subtracted terms by

$$(q_{i+2}(2) - q_{i+2}(1)) + (q_i(4) - q_i(3)) = \log_2\left(\frac{8q_m m_1 h_m}{\varepsilon^3 \delta}\right) + \log_2\left(\frac{8\ln(Nh_m/\delta)}{\varepsilon^3}\right)$$
$$= \log_2\left(\frac{64 m_1 \ln(Nh_m/\delta) q_m h_m}{\varepsilon^7 \delta}\right).$$

By Lemma C.2 we have that $q_{i+1}(3) - q_{i+1}(2) \geq \log_2\left(\frac{N\varepsilon^5}{32m_1\mu q_m}\right)$. Thus, combining both equations we get that

$$q_{(i+2)}(1) - q_i(4) = \log_2\left(\frac{N\varepsilon^5}{32m_1\mu q_m}\right) - \log_2\left(\frac{64m_1\ln(Nh_m/\delta)q_m h_m}{\varepsilon^7\delta}\right)$$
$$= \log_2\left(\frac{N\varepsilon^{12}\delta}{2048m_1^2\mu\ln(Nh_m/\delta)q_m^2 h_m}\right).$$

If $N \geq \frac{2048m_1\mu\ln(Nh_m/\delta)q_m^2 h_m}{\varepsilon^{12}\delta}$ then we have $q_{(i+2)}(1) - q_i(4) \geq 0$ and thus by Lemma C.10, the expected contribution of any weight class $W_q^+$ is at most $2\|W_q^+\|_1$.   □

If $N < \frac{2048m_1^2\mu\ln(Nh_m/\delta)q_m^2 h_m}{\varepsilon^{12}\delta}$ we have the following adaptation of the lemma:

**Lemma C.12.** *If for some $k \in \mathbb{N}$ we choose $N_i = N \geq \frac{32m_1\mu q_m}{\varepsilon^5} \cdot \left(\frac{64m_1\ln(Nh_m/\delta)q_m h_m}{\varepsilon^7\delta}\right)^{1/(k-1)}$ for all $i \in [h_m]$, and $M_i$ solving the equation $q_{(M_{i-1},N)}(3) = q_{(M_i,N)}(2)$, then the expected contribution of any weight class $W_q^+$ is at most $k\|W_q^+\|_1$.*

*Proof.* We generalize the proof of Lemma C.11. We can substitute $q_{(i+k)}(1) - q_i(4)$ as follows:

$$q_{(i+k)}(1) - q_i(4) = q_{(i+k)}(1) - q_{(i+k)}(2) + q_i(3) - q_i(4) + q_{(i+k)}(2) - q_i(3)$$
$$= q_{(i+k)}(2) - q_i(3) - (q_{(i+k)}(2) - q_{(i+k)}(1)) - (q_i(4) - q_i(3))$$
$$= q_{(i+k-1)}(3) - q_{i+1}(2) - (q_{(i+k)}(2) - q_{(i+k)}(1)) - (q_i(4) - q_i(3))$$
$$= \sum_{j=1}^{k-1} q_{(i+j)}(3) - q_{i+j}(2) - (q_{(i+k)}(2) - q_{(i+k)}(1)) - (q_i(4) - q_i(3)).$$

The difference to the proof of Lemma C.11 is the telescoping sum. We have that

$$\sum_{j=1}^{k-1} q_{(i+j)}(3) - q_{i+j}(2) = (k-1)\cdot\log_2\left(\frac{N\varepsilon^5}{32m_1\mu q_m}\right) = \log_2\left(\left(\frac{N\varepsilon^5}{32m_1\mu q_m}\right)^{k-1}\right).$$

Thus if $N \geq \frac{32m_1\mu q_m}{\varepsilon^5} \cdot \left(\frac{64m_1\ln(Nh_m/\delta)q_m h_m}{\varepsilon^7\delta}\right)^{1/(k-1)}$ we have that $\sum_{j=1}^{k-1} q_{(i+j)}(3) - q_{i+j}(2) \geq \log_2\left(\frac{64m_1\ln(Nh_m/\delta)q_m h_m}{\varepsilon^7\delta}\right)$.

Further note that $(q_{i+2}(2) - q_{i+2}(1)) + (q_i(4) - q_i(3)) = \log_2\left(\frac{64m_1\ln(Nh_m/\delta)q_m h_m}{\varepsilon^7\delta}\right)$ as before. We conclude that $q_{(i+k)}(1) - q_i(4) > 0$. Consequently, applying Lemma C.10 finishes the proof.   □

Next we want to show how we can reduce the expected contribution of all weight classes below $2\|W_q^+\|_1$. To this end we first increase the number of buckets at each level so as to get

$$\log_2\left(\frac{N\varepsilon^5}{32m_1\mu q_m}\right) \geq k\log_2\left(\frac{64m_1\ln(Nh_m/\delta)q_m h_m}{\varepsilon^7\delta}\right).$$

Note that the expected contribution of any important weight class $W_q^+$ is at least $\|W_q^+\|_1$. Moreover, the above choice ensures that all but a $k$-th fraction of weight classes have an expected contribution of exactly $\|W_q^+\|_1$, and only the remaining $k$-th fraction has a larger expected contribution that crucially is still bounded by $2\|W_q^+\|_1$. Then the last step is to add a random shift so that the probability of each weight class $W_q^+$ for having an expected contribution of $2\|W_q^+\|_1$ is at most $\frac{1}{k}$. To simplify notation we set $N_1' = \frac{32m_1\mu q_m}{\varepsilon^5}$ and $N_2' = \frac{64m_1\ln(n)q_m h_m}{\varepsilon^7\delta}$ and assume that $n \geq N^k h_m/\delta$.

**Lemma C.13.** *Let $\gamma = \frac{1}{k} < 1$ for some $k \in \mathbb{N}$. Assume that $N_0$ is chosen uniformly at random from $N^{(1)},\dots N^{(1/\gamma)}$ where $N^{(i)} = N_0' \cdot N_2'^i$. Further let $N_i = N = N_1' \cdot N_2'^{k+1}$ for any $i > 0$. Then the expected contribution of any weight class $W_q^+$ is at most $(1+\gamma)\|W_q^+\|_1$.*

*Proof.* First note that

$$\log_2\left(\frac{N\varepsilon^5}{32m_1\mu q_m}\right) - k\log_2\left(\frac{64m_1\mu\ln(n)q_m h_m}{\varepsilon^7\delta}\right) = \log_2(N/N_1') - \log_2(N_2'^k) \geq 0.$$

This shows that the relation of weight classes that are relevant on two levels to the weight classes that are relevant on only one level is $1:k$. By choosing $N_0$ at random we introduce a shift by $i\log_2(N_2')$, which is the maximal length of a block $[q_{i-1}(1), q_i(4)]$. Hence, for each $q \in \mathbb{N}$ there can be only one $i$ such that $q$ is relevant in two levels. This implies that the expected contribution of $W_q^+$ is at most $\frac{k-1}{k}\cdot\|W_q^+\|_1 + \frac{1}{k}\cdot 2\|W_q^+\|_1 = (1+\frac{1}{k})\|W_q^+\|_1$. □

## C.5 NET ARGUMENT

To get a weak weighted sketch we need the contraction bounds not just for a single solution but for all $\beta \in \mathbb{R}^d$. For now we ignore the variance regularization and focus only on $f_1$, i.e., on plain logistic regression. We first show that if the distance of two vectors $v, v' \in \mathbb{R}^n$ is small then $|f_1(v) - f_1(v')|$ is also small.

**Lemma C.14.** *For any $v, v' \in \mathbb{R}^n$ with $\|v - v'\|_1 \leq \varepsilon$ it holds that $|f_1(v) - f_1(v')| \leq \varepsilon$.*

*Proof.* Since $\ell'(v) = \frac{e^v}{e^v+1} \leq 1$ we get that

$$|f_1(v) - f_1(v')| \leq \sum_{i=1}^n |\ell(v_i) - \ell(v_i')| \leq \sum_{i=1}^n |v_i - v_i'| = \|v - v'\|_1$$

which proves the lemma. □

**Lemma C.15.** *Assume that for $\beta \in \mathbb{R}^d$ it holds that $|f_1(X'\beta) - f_1(X\beta)| \leq \varepsilon$. Then for any $\beta' \in \mathbb{R}^d$ with $\|X\beta - X\beta'\|_1 \leq \varepsilon/(b^{h_m}h_m)$ it holds that $|f_1(X\beta') - f_1(X'\beta')| \leq 3\varepsilon$.*

*Proof.* It holds that $\|X'(\beta - \beta')\|_1 = \|SX(\beta - \beta')\|_1 \leq b^{h_m}h_m\|X(\beta - \beta')\|_1 \leq \varepsilon$ since for each $i \in [n]$ there are at most $h_m$ columns $j$ such that $S_{ij} \neq 0$ and each entry of $S$ is bounded by $b^{h_m}$. Thus, using the triangle inequality and applying Lemma C.14 yields

$$|f_1(X\beta') - f_1(X'\beta')| \leq |f_1(X'\beta') - f_1(X'\beta)| + |f_1(X'\beta) - f_1(X\beta)| + |f_1(X\beta) - f_1(X\beta')|$$
$$\leq \varepsilon + \varepsilon + \varepsilon \leq 3\varepsilon.$$

□

**Lemma C.16.** *There exists a net $\mathcal{N} \subset \mathbb{R}^d$ of size $|\mathcal{N}| = \exp\left(\mathcal{O}(d\ln(n))\right)$ such that for any point $y \in \mathbb{R}^d$ with $\|Xy\|_1 \leq n\mu$ there exists a point $y' \in \mathcal{N}$ such that $\|Xy' - Xy\|_1 \leq \frac{\varepsilon}{\mu b^{h_{\max}}h_m}$.*

*Proof.* We set

$$\mathcal{N} = \left\{\beta = v\cdot\frac{\varepsilon}{db^{h_m}h_m} \mid v \in \mathbb{Z}^d \text{ with } \|v\|_\infty \leq \frac{dn\mu b^{h_{\max}}h_m}{\varepsilon}\right\}. \tag{6}$$

Then for any $y \in \mathbb{R}$ with $\|Xy\|_1 \leq n\mu$ the point $Xy' = \lfloor\frac{db^{h_m}h_m}{\varepsilon}\cdot Xy\rfloor\cdot\frac{\varepsilon}{db^{h_m}h_m}$ is in $\mathcal{N}$ and it holds that $\|Xy - Xy'\|_1 \leq d\cdot\frac{\varepsilon}{db^{h_m}h_m} = \frac{\varepsilon}{b^{h_m}h_m}$. Further we have $|\mathcal{N}| \leq \left(\frac{d^2n\mu b^{2h_m}h_m^2}{\varepsilon^2}\right)^d = \exp\left(\mathcal{O}(d\ln(n))\right)$. □

Combining Lemma C.15 and Lemma C.16 we get:

**Lemma C.17.** *There exists a net $\mathcal{N} \subset \mathbb{R}^d$ with $|\mathcal{N}| = \exp\left(\mathcal{O}(d\ln(n))\right)$ such that if $|f_1(X'\beta) - f_1(X\beta)| \leq \varepsilon$ holds for any $\beta \in \mathcal{N}$, then for any $\beta' \in \mathbb{R}^d$ with $\|X\beta'\|_1 \leq n\mu$ it holds that $|f_1(X'\beta') - f_1(X\beta')| \leq 3\varepsilon$.*

### C.6 Constant factor approximation changes

To prove the first part of Theorem 2 we need one more tweak in level 0. Since we only aim to achieve a constant factor approximation with constant probability we can assume that $\varepsilon$ and $\delta$ are constant.

**Heavy hitters - alternative version**

There is another way of handling heavy hitters. Using it we can reduce the sketch size at the cost of running time. The idea is that each row gets sampled multiple times. More precisely, we replace level 0 by the following sketch: At level 0 we map each element to $s = 8m_1 q_m \mu / \varepsilon^2$ rows. Technically we are getting rid of heavy hitters this way. To compensate the fact that each element appears multiples times, we set the weight of buckets of level 0 to $w_0 = 1/s$.

### C.7 Proof of Theorem 2

We are now ready to prove Theorem 2:

*Proof of Theorem 2.* If $\beta = 0$ is a $1 - 2\varepsilon$ approximation, then we get the dilation bounds for free since $f_{1w}(X'\beta) = \ln(2) = f_1(X\beta)$. Otherwise let $\beta^*$ be the minimizer of $f_1(X\beta)$. Note that $\beta^*$ satisfies the assumption of Lemma C.9.

1) We fix constants $\varepsilon = 1/8$ and $\delta = 1/8$.

We use the alternative approach for handling heavy hitters and define $M_i$ and $N_i$ as in Lemma C.12 for some constant $k = 1 + \frac{1}{c}$ and set $h_m = \min\{i \mid M_i \le N\}$.

By Lemma C.12 the expected contribution of any weight class is at most $k\|W_q^+\|_1$. Thus using Markov's inequality we can bound $f_{1w}(SX\beta^*) \le akf_1(X\beta^*)$ with probability $\frac{1}{a}$ for any $a \in \mathbb{N}$. In other words, it is constant with constant probability. By our choice of $M_i$ and $N_i$, the contraction bounds hold for any $X\beta$ with failure probability at most $(2h_m + 2q_m + 2)e^{-m_1}$ by combining Lemma C.8 and Lemma B.3. Setting $m_1 = \mathcal{O}(d \ln(n))$ and using Lemma C.17 we get that the contraction bounds hold for all $\beta \in \mathbb{R}^d$ with $\|X\beta\|_1 \le n\mu$. We note that the contraction bounds can be extended to any $\beta \in \mathbb{R}^d$ since $f_1(X\beta) \approx \|X\beta\|_1$ if $\|X\beta\|_1 > n\mu$. We refer to (Munteanu et al., 2021) for details. Further note that $q_i(2) < q_i(3)$, and thus $h_m \le \log_2(2^{q_m}) = O(\ln(n))$. The number of buckets at each level is $N = \frac{32m_1\mu q_m}{(1/8)^5} \cdot \left(\frac{64m_1 \ln(Nh_m/\delta)q_m h_m}{(1/8)^7}\right)^c$. We specify the number $r$ of rows of $SX$, which is $r = h_m N$. Since $h_m, q_m = O(\ln(n))$ and $m_1 = \mathcal{O}(d \ln(n))$ we get that $r = O(\mu d^{1+c} \ln(n)^{2+4c})$. The running time of our algorithm is $O(\mu d \ln(n)\mathtt{nnz}(X))$ since each row $x_i$ gets assigned to $O(\mu d \ln(n))$ buckets.

2') Before proving the second part we show that with $r = O(\frac{\mu^2 d^4 \ln(n)^7}{\varepsilon^{12}\delta})$ and $T = O(\mathtt{nnz}(X))$ we can get an approximation factor of $\alpha = 1 + (1+\varepsilon)a$ and failure probability of $P = \delta + \frac{1}{a}$. There are only a few differences compared to the proof of the first part: instead of Lemma C.12 we use Lemma C.11. Hence we need the number of buckets to be

$$N = \max\left\{N_0', \frac{2048m_1^2\mu^2 \ln(Nh_m/\delta)q_m^2 h_m}{\varepsilon^{12}\delta}\right\} = \frac{2048m_1^2\mu^2 \ln(Nh_m/\delta)q_m^2 h_m}{\varepsilon^{12}\delta}.$$

Consequently we have that $r = h_m N = O(\frac{\mu^2 d^4 \ln(n)^7}{\varepsilon^{12}\delta})$. Since every row gets assigned to $O(1)$ buckets the running time is $O(\mathtt{nnz}(X))$. Now assume that the contraction bound holds for $\beta^*$. Then $Y = f_{1w}(SX\beta^*) - (1-\varepsilon)f_1(X\beta^*)$ is a positive random variable with expected value at most $(1+\varepsilon)f_1(X\beta^*)$, and thus using Markov's inequality gives us that $Y > a(1+\varepsilon)f_1(X\beta^*)$ holds with probability at most $\frac{1}{a}$. Hence it follows that $f_{1w}(SX\beta^*) \le f_1(X\beta^*) + a(1+\varepsilon)f_1(X\beta^*)$ with failure probability at most $\frac{1}{a}$.

2) The proof is again similar to 2'). The only difference is that we use Lemma C.13 instead of Lemma C.11. Hence the number of buckets at each level is bounded by $N = \max\{N_0', N_1' \cdot N_2'^{1+\varepsilon^{-1}}\}$. Thus $r = h_m N = O(\frac{d^2 h_m q_m^2 \mu^2 m_1^2}{\delta\varepsilon^7} + \frac{32d\mu \ln(n)^2}{\varepsilon^5} \cdot \left(\frac{64d \ln(n)^4}{\varepsilon^7\delta}\right)^{1+\varepsilon^{-1}})$. $\qquad\square$

# D   APPROXIMATING $\|X\beta - Y\|_1$

The sketching algorithm is the same as before and also the analysis is very similar to the previous part. We start with a fixed point $z = (X, -Y)\beta'$, where $\beta' = (\beta, 1) \in \mathbb{R}^d$ and analyze $Sz$. Again we assume that $\|z\|_1 = 1$. Instead of weight classes $W_q^+$ we use weight classes $W_q = \{i \in [n] \mid |z_i| \in (2^{-q-1}, 2^{-q}]\}$. Since we are only dealing with absolute values, which are symmetric, we no longer need to parameterize by $\mu$. We can continue to use the same definitions for $q_h(1)$, $q_h(2)$ and $q_h(3)$ when setting $\mu$ in those bounds to be 1. We will only slightly change $q_h(3)$ since we will need another trick to prove the second outer bound $q_h'(4)$.

## D.1   DILATION BOUNDS FOR $\ell_1$

For approximating $\ell_1$ we need a different approach for $q_h(4)$ when bounding the contribution of small entries at each level. The idea is to use a Ky-Fan norm argument to remove the smallest contributions from the $\ell_1$-norm. At a fixed level $h$ we put $\mathcal{B}_h$ to be the set of buckets at level $h$ and $\mathcal{B}_h'$ to be the set of buckets with the $p2^{q_h(3)} \leq \frac{\varepsilon N}{h_m}$ largest entries with respect to $|G(B)|$ where $q_h(3) := \ln(\min\{\frac{\varepsilon N}{2h_m}, \frac{N\varepsilon^2}{4p}\})$. We further define

$$K(h) = \sum_{B \in \mathcal{B}_h'} |G(B)|.$$

Since $\bigcup_{q \in [q_h(2), q_h'(3)]} W_q^*$ contains at most $p2^{q_h'(3)}$ elements, we have that $K(h) \geq \|W_q^*\|_1$. We set $q_h'(4) = \ln(\frac{3Nh_m \ln Nh_m/\delta}{p\varepsilon})$. Set $Y_2 = Y_2(h) = \{i \in [n] \mid |z_i| \leq \gamma_2 := \frac{cp}{N \ln(Nh_m/\delta)}\}$.

**Lemma D.1.** *With failure probability at most $\frac{\delta}{h_m N}$ it holds that for any bucket $B$ at level $h$ we have that*

$$\sum_{i \in B \cap Y_2} |z_i| \leq \max\{2 \cdot \frac{p \cdot \|Y_2\|_1}{N}, \frac{p}{N}\left(\|Y_2\|_1 + \frac{\varepsilon}{h_m}\right)\}$$

*Proof.* Fix a bucket $B$ at level $h$. For $i \in Y_2$ let $X_i = z_i$ if $i \in B$ and $X_i = 0$ otherwise. Then we have $E := \mathbb{E}(\sum_{i \in Y_2} X_i) = \frac{p \cdot \|Y_2\|_1}{N}$. Further we have $\mathbb{E}(\sum_{i \in Y_2} X_i^2) = \sum_{i \in Y_2} \frac{p}{N} \cdot z_i^2 \leq \frac{\gamma_2 p}{N} \cdot \sum_{i \in Y_2} |z_i| = \gamma_2 E$. We set $\lambda = \max\{E, \frac{\varepsilon}{Nh_m}\}$ Then using Bernstein's inequality we get that

$$P(\sum_{i \in Y_2} X_i \geq E + \lambda) \leq \exp\left(\frac{-\lambda^2/2}{\gamma_2 E + \gamma_2 E/3}\right)$$

$$\leq \exp\left(\frac{-\lambda^2/2}{\gamma_2 \lambda + \gamma_2 \lambda/3}\right)$$

$$\leq \exp\left(\frac{-\lambda}{3\gamma_2}\right)$$

$$\leq \exp\left(\frac{-p\varepsilon}{3Nh_m\gamma_2}\right)$$

$$\leq \exp\left(-\ln(Nh_m/\delta)\right) \leq \frac{\delta}{h_m N}.$$

$\square$

**Lemma D.2.** *With failure probability at most $\delta$ it holds that*

$$\sum_{h \leq h_m} \sum_{i \in Y_2(h) \cap \bigcup_{B \in \mathcal{B}_h'} B} z_i \leq \varepsilon$$

*Proof.* Using the union bound over the event from Lemma D.1 over all $Nh_m$ buckets, using that $|\mathcal{B}_h'| \leq \varepsilon N/2h_m$ and $\max\{2 \cdot \|Y_2\|_1, \left(\|Y_2\|_1 + \frac{\varepsilon}{h_m}\right)\} \leq 2$ we get that

$$\sum_{i \in Y_2(h) \cap \bigcup_{B \in \mathcal{B}_h'} B} z_i \leq \frac{\varepsilon N}{2h_m} \cdot \frac{2p}{N} \leq \frac{\varepsilon}{h_m}.$$

holds for every level $h$ with failure probability at most $\delta$. Summing up over all levels we get $\sum_{h \in h_m} \sum_{i \in Y_2(h) \cap \bigcup_{B \in \mathcal{B}'_h} B} z_i \leq \varepsilon$. $\qquad \square$

We have the following lemmas using similar proofs as in the previous section:

**Lemma D.3.** *If for some $k \in \mathbb{N}$ we choose $N_i = N \geq \frac{32 m_1 q_m h_m}{\varepsilon^5} \cdot \left( \frac{64 m_1 \ln(N h_m / \delta) q_m h_m^2}{\varepsilon^6 \delta} \right)^{1/(k-1)}$ for all $i \in [h_m]$ and $M_i$ solving the equation $q_{(M_{i-1}, N)}(3) = q_{(M_i, N)}(2)$, then the expected contribution of any weight class $W_q$ is at most $(k + \varepsilon) \|W_q\|_1$.*

Here the additional $\varepsilon$ comes from Lemma D.1.

We set $N_0'' = N_0'$, $N_1'' = \frac{32 m_1 q_m h_m}{\varepsilon^5}$ and $N_2'' = \frac{64 m_1 \ln(N h_m / \delta) q_m h_m^2}{\varepsilon^6 \delta}$ and assume that $n \geq N^k h_m / \delta$.

**Lemma D.4.** *Let $\gamma = \frac{1}{k} < 1$ for some $k \in \mathbb{N}$. Assume that $N_0$ is chosen uniformly at random from $N^{(1)}, \ldots N^{(1/\gamma)}$ where $N^{(i)} = N_0'' \cdot N_2''^i$. Further let $N_i = N = N_1'' \cdot N_2''^{k+1}$ for any $i > 0$. Then the expected contribution of any weight class $W_q$ is at most $(1 + \gamma) \|W_q\|_1$.*

## D.2 NET ARGUMENT

For $\beta \in \mathbb{R}^{d+1}$ we set $g_1(\beta) = \|(X, -Y)\beta)\|_1$ and $g_2(\beta) = \|(SX, -SY)\beta)\|_1$

**Lemma D.5.** *Assume that for $\beta \in \mathbb{R}^{d+1}$ it holds that $|g_1(\beta) - g_2(\beta)| \leq \varepsilon$. Then for any $\beta' \in \mathbb{R}^d$ with $\|X\beta - X\beta'\|_1 \leq \varepsilon / (b^{h_m} h_m)$ it holds that $|g_1(\beta') - g_2(\beta')| \leq 3\varepsilon$.*

*Proof.* It holds that $\|X'(\beta - \beta')\|_1 = \|SX(\beta - \beta')\|_1 \leq b^{h_m} h_m \|X(\beta - \beta')\|_1 \leq \varepsilon$ since for each $i \in [n]$ there are at most $h_m$ columns $j$ such that $S_{ij} \neq 0$ and each entry of $S$ is bounded by $b^{h_m}$. Also note that $\|g_i(v) - g_i(v')\|_1 \leq \|v - v'\|_1$ holds for any two vectors $v, v' \in \mathbb{R}^{d+1}$. Thus, using the triangle inequality yields

$$|g_1(\beta') - g_2(\beta')| \leq |g_2(\beta') - g_2(\beta)| + |g_2(\beta) - g_1(\beta)| + |g_1(\beta) - g_1(\beta')|$$
$$\leq \varepsilon + \varepsilon + \varepsilon \leq 3\varepsilon.$$

$\qquad \square$

**Lemma D.6.** *There exists a net $\mathcal{N} \subset \mathbb{R}^d$ with $|\mathcal{N}| = \exp\left(\mathcal{O}(d \ln(n))\right)$ such that if $|g_1(\beta) - g_2(\beta)| \leq \varepsilon g_1(\beta)$ holds for any $\beta \in \mathcal{N}$ then for any $\beta' \in \mathbb{R}^{d+1}$ it holds that $|g_1(\beta') - g_2(\beta')| \leq 3\varepsilon g_1(\beta')$.*

*Proof.* We set

$$\mathcal{N} = \left\{ \beta = v \cdot \frac{\varepsilon}{d b^{h_m} h_m} \mid v \in \mathbb{Z}^d \text{ with } \|v\|_\infty \leq \frac{d b^{h_m} h_m}{\varepsilon} \right\}. \tag{7}$$

Then it holds that for any $\beta \in \mathbb{R}^{d+1}$ with $g_1(\beta) = 1$ the point $(X, -y)\beta' = \lfloor \frac{d b^{h_m} h_m}{\varepsilon} \cdot (X, -y)\beta) \rfloor \cdot \frac{\varepsilon}{d b^{h_m} h_m}$ is in $\mathcal{N}$ and it holds that $\|(X, -y)\beta'\|_1 \leq d \cdot \frac{\varepsilon}{d b^{h_m} h_m} = \frac{\varepsilon}{b^{h_m} h_m}$. Using Lemma D.5 it holds that $|g_1(\beta') - g_2(\beta')| \leq 3\varepsilon \leq 3\varepsilon g_1(\beta)$. Further we have $|\mathcal{N}| \leq \left( \frac{d b^{h_m} h_m}{\varepsilon} \right)^{2d} = \exp\left(\mathcal{O}(d \ln(n))\right)$. Now for any $r \in \mathbb{R}$ and $\beta \in \mathbb{R}^{d+1}$ with $g_1(\beta) = 1$ we have that $|g_1(r\beta) - g_2(r\beta)| = |r g_1(\beta) - r g_2(\beta)| = r |g_1(\beta) - g_2(\beta)| \leq 3\varepsilon r$. $\qquad \square$

## E SKETCHING VARIANCE-BASED REGULARIZED LOGISTIC REGRESSION

In this section we show that our algorithm also approximates the variance well under the assumption that roughly $f_1(X\beta) \leq \ln(2)$. We stress that this assumption does not rule out the existence of good approximations. Indeed, even the minimizer is contained as observed in the preliminaries, since we have that $\min_{\beta \in \mathbb{R}^d} f(X\beta) \leq f(0) = f_1(0) = \ln(2)$. Again we focus on a single $z = X\beta$ first. What remains to show is that $\sum_{i: z_i > 0} z_i^2$ is approximated well. We set $H(z) = \sum_{i=1}^n z_i^2$, $H^+(z) = \sum_{i: z_i > 0} z_i^2$ and $h(y) = \frac{y^2}{H^+(z)}$. By $\mu$-complexity we get that $H^+(z) \geq \frac{H(z)}{\mu}$. We define $W_q^2 = \{i \in [n] \mid h(z_i) \in (2^{-q-1}, 2^q]\}$ and $W_q^1 = \{i \in [n] \mid \frac{z_i}{\|z\|_1} \in (2^{-q-1}, 2^q]\}$. As the argument

is almost the same as in the section before, we will only note the differences. We will also use the same definition of importance, i.e., a weight class $W_q^2$ is important if $H^+(W_q^2) \geq \frac{\varepsilon}{q_m \mu}$. Similar to the previous analysis we have that if $W_q^2$ is important then $|W_q^2| \geq \frac{\varepsilon 2^q}{q_m \mu}$. With those adapted definitions we proceed by adapting the main lemmas of Section C that finally yield Theorem 3.

**Lemma E.1.** *For any $z_i \in W_q^2$ there exists $q' \leq (q-1)/2 + \ln(n)/2$ such that $z_i \in W_{q'}^1$.*

*Proof.* It is well known that $\|z\|_1 \leq \sqrt{n}\|z\|_2$. We conclude that

$$\frac{z_i}{\|z\|_1} \geq \frac{z_i}{\sqrt{n}\|z\|_2} = \frac{1}{\sqrt{n}} \frac{z_i^2}{\|z\|_2^2} \bigg/ \sqrt{\frac{z_i^2}{\|z\|_2^2}} \geq \frac{1}{\sqrt{n}} \cdot \frac{2^{-q-1}}{2^{-(q-1)/2}}.$$

Now taking the logarithm proves the lemma. $\qquad\square$

**Contraction bounds** Recall that:

$$\gamma_1 := \frac{p}{3m_1}$$
$$Y_1 := \{i \in [n] \mid |z_i| \geq \gamma_1\}$$

Here $Y_1$ is the set of 'large elements'. We redefine $\mu_z = \frac{\sum_{z_i > 0} z_i^2}{\sum_{z_i < 0} z_i^2}$

**Lemma E.2.** *The following hold:*

1) *$|Y_1 \cap U| \leq \varepsilon N/2$ with failure probability at most $\exp(-m_1)$;*

2) *Let $\mathcal{B} = \{B \in \mathcal{B}_h \mid \sum_{i \in B \setminus Y_1} |z_i| \leq \frac{4p}{\varepsilon N}\}$. Then $|\mathcal{B}| \geq (1 - \varepsilon/2)N$ with failure probability at most $\exp(-m_1)$;*

3) *Assume that $q \geq \log_2(\frac{8 q_m \mu_z m_1}{\varepsilon^3 p}))$ and that $W_q^2$ is important or that $|W_q| \geq 8m_1 \varepsilon^{-2} \cdot p^{-1}$. Then with failure probability at most $\exp(-m_1)$ there exists $W_q^* \subset W_q^2 \cap \mathcal{B}$ such that $\|W_q^*\|_1 \geq (1-\varepsilon)^2 \|W_q^+\|_1 \cdot p$ and each element of $W_q^*$ is in a bucket in $\mathcal{B}$ containing no other element of $Y_1$;*

4) *If $q \leq \log_2(\frac{N\varepsilon^2}{\sqrt{n}4p})$ and $W_q^*$ as in 3) exists, then with failure probability at most $\exp(-m_1)$ it holds that $\sum_{i \in W_q^*} G(B_i) \geq (1-\varepsilon)\|W_q^*\|_1$.*

The proof is verbatim to the proof of Lemma C.3. For the 4th part we use Lemma E.1 to reduce the problem to the weight class $W_q^1$. This causes an additional term of $\frac{1}{\sqrt{n}}$ in the logarithm of $q_3(M, N)$.

We also have a change in $q_4(M, N)$. More precisely we need two additional factors of $\varepsilon$ in $\gamma_2$:

$$\gamma_2 := \frac{M\varepsilon^4}{2Nn\ln(Nh_{\max}/\delta)}$$
$$Y_2 = \{i \in [n] \mid |z_i| \leq \gamma_2\}: \text{Set of small elements};$$
$$Y_2^+ = \{i \in [n] \mid |z_i| \leq \gamma_2, z_i \leq 0\}: \text{Set of small negative elements};$$
$$Y_2^- = \{i \in [n] \mid z_i \leq \gamma_2, z_i \geq 0\}: \text{Set of small positive elements};$$

Further we set $A := \sum_{z_i \geq 0} z_i$, $A' = \sum_{z_i \in Y_2^-} |z_i|$, $A_1 = \sum_{z_i \in Y_2^+} |z_i|$ and $A_2 = A - A_1 \geq 0$.

**Lemma E.3.** *If $A' \geq A(1+\varepsilon)$ then for any bucket $B$ that contains only elements of $Y_2$ we have that $G(B) = \sum_{i \in B} z_i \leq \frac{M}{Nn} \cdot (-A_2)$ with failure probability at most $\frac{\delta}{Nh_{\max}}$.*

*Proof.* Let $X_i$ be the random variable attaining value $z_i$ if $i \in B$ and 0 otherwise, for $i \in [n]$. The expected value for $G(B) = \sum_{i \in [n]} X_i$ is $E' := \frac{M}{nN} \cdot (A' - A_1)$. Further we have that

$$\mathbb{E}\left(\sum_{i \in [n]} X_i^2\right) = \sum_{i \in Y_2} \frac{M}{nN} \cdot z_i^2 \leq \frac{M}{nN} \cdot \sum_{i \in Y_2} \gamma_2 z_i \leq \frac{\gamma_2 M}{nN}$$

since all $X_i$ are bounded by $\gamma_2$ by assumption. Applying Bernstein's inequality thus yields

$$P(G(B) > 0) \leq P\left(\sum_{i \in [n]} X_i - E' \geq \varepsilon |E'|\right) \leq \exp\left(\frac{-\varepsilon^2 |E'|^2/2}{\gamma_2 \cdot M/(nN) + \varepsilon \gamma_2 |E'|/3}\right)$$

$$\leq \exp\left(\frac{-\varepsilon^3 \cdot M/(nN)/2}{\gamma_2(M/(nNE') + \varepsilon/3)}\right)$$

$$= \exp\left(\frac{-\varepsilon^3 \cdot M/(nN)/2}{\gamma_2 \varepsilon^{-1}((A' - A_1) + 1/3)}\right)$$

$$\leq \exp\left(\frac{-\varepsilon^4 \cdot M/(nN)}{2\gamma_2}\right)$$

$$\leq \exp\left(-\ln\left(\frac{Nh_{\max}}{\delta}\right)\right)$$

$$= \frac{\delta}{Nh_{\max}}.$$

Note that $\varepsilon E' \leq \varepsilon \cdot \frac{M}{nN} \cdot (A' - A_1) \leq \varepsilon \cdot \frac{M}{nN} \cdot A$ and thus $\mathbb{E}(\sum_{i \in [n]} X_i^2) + \varepsilon E' \leq \frac{M}{nN} \cdot (-A' + A_1 + \varepsilon A) \leq \frac{M}{nN} \cdot (-A_2)$. □

Our main lemma thus changes to:

**Lemma E.4.** *With probability at least* $1 - \frac{\delta}{h_m}$ *the weight classes* $W_q^2$ *for* $q \geq q_{(M,N)}(4) := \log_2(\gamma_2^{-1}) := \log_2(\frac{2Nn \ln(Nh_m/\delta)}{M\varepsilon^4})$ *and* $q \leq q_{(M,N)}(1) := \log_2(\frac{n\delta}{Mh_m})$ *have zero contribution to* $\sum_B G^+(B)$*, i.e., for any bucket* $B$ *we have* $\sum_{z_i \in B \setminus I_r} z_i \leq 0$ *where* $I_r = \{i \in [n] \mid z_i \in W_q, q \in [q_{(M,N)}(1), q_{(M,N)}(4)]\}$*. Further, with failure probability at most* $\exp(-m_1)$*, for each* $\log_2(\frac{8q_m \mu m_1 n}{\varepsilon^3 M})) =: q_{(M,N)}(2) \leq q \leq q_{(M,N)}(3) := \log_2(\frac{Nn\varepsilon^2}{4Mm_1\sqrt{n}})$ *there exists* $W_q^*$ *such that* $\sum_{i \in W_q^*} G(B_i) \geq (1 - \varepsilon)^2 \|W_q^2\|_2 \cdot \frac{M}{n}$*. Thus it holds that:*

$$q_{(M,N)}(2) - q_{(M,N)}(1) = \log_2\left(\frac{8q_m m_1 h_m}{\varepsilon^3 \delta}\right)$$

$$q_{(M,N)}(3) - q_{(M,N)}(2) = \log_2\left(\frac{N\varepsilon^5}{32m_1 \mu q_m \sqrt{n}}\right) =: \log_2(b)$$

$$q_{(M,N)}(4) - q_{(M,N)}(3) = \log_2\left(\frac{8\ln(Nh_m/\delta\sqrt{n})}{\varepsilon^6}\right).$$

If $N = M$ then we set $q_{(M,N)}(3) = q_{(M,N)}(4) = \infty$. If $M = n$ then we set set $q_{(M,N)}(1) = q_{(M,N)}(2) = 0$. We set $q_h(i) = q_{(M_h,N_h)}(i)$ for $i \in \{1,2,3,4\}$ and $Q_h = [q_h(2), q_h(3)]$ to be the well-approximated weight classes and $R_h = [q_h(1), q_h(4)]$ to be the relevant weight classes at level $h$. Note that $q_{(M,N)}(1)$ and $q_{(M,N)}(2)$ stay the same as before.

**Heavy hitters** The important changes to note here are that we need to replace Lemma C.5 with an appropriate lemma for the $\ell_2$-leverage scores and there is an additional factor of $\frac{1}{\sqrt{n}}$ in $\gamma_4$. We further redefine $u_p$ to be the $\ell_2$-leverage scores.

**Lemma E.5.** *(Clarkson & Woodruff, 2015a) If* $u_i$ *is the* $k$*-th largest* $\ell_2$*-leverage score, then for* $z$ *in the subspace spanned by the columns of A it holds that* $z_i^2 \leq \frac{d}{k} \sum_{j=1}^n z_j^2$*. Further it holds that* $\sum_{i=1}^n u_i = d$

The lemma follows as in the case of $\ell_1$ leverage scores by using an orthonormal basis.

We then apply Lemma C.6 and Lemma E.5 as before: set $N_1 = d\gamma_3^{-1}$ and $N_2 = d\gamma_3^{-1} \cdot \gamma_4^{-1}$, where $\gamma_3 = \frac{\varepsilon^3}{8q_m \mu m_1}$ and $\gamma_4 = \frac{2\varepsilon}{\sqrt{n}m_1}$. Further let $Y_3$ (resp. $Y_4$) be the set of coordinates with the $N_1$ (resp. $N_2$) largest leverage scores. We denote by $\mathcal{E}_2$ the event that all coordinates in $Y_3$ are in a bucket with no other member of $Y_4$. By Lemma C.6, $\mathcal{E}_2$ holds with probability at least $1 - \delta$ for an appropriate

$N = N_0^{(2)} = N_1 N_2 \delta^{-1} = \mathcal{O}(\frac{d^2 q_m^2 \mu^2 m_1^3 \sqrt{n}}{\delta \varepsilon^7})$. For any entry $z_p \in H$ we have $z_p \geq \gamma_3$ and thus by Lemma C.5, we have $p \in Y_3$ and for any entry $p \notin Y_4$ we have $z_p < \gamma_3 \cdot \gamma_4$. It remains to show that the remaining entries in the buckets containing a heavy hitter only have a small contribution. To this end we use Bernstein's inequality. For a coordinate $p \in [n]$ we denote by $B_p$ the bucket at level 0 that contains $p$.

**Lemma E.6.** *Assume $\mathcal{E}_2$ holds. Then for any $z_i \in H$ we have $G(B_i) \geq (1 - \varepsilon)z_i$.*

### Contraction bounds for a single point

**Lemma E.7.** *Assume that $\mathcal{E}_2$ holds. Denote by $z_i'$ the $i$-th row of $SX\beta$ for $i \in n'$. Then with failure probability at most $(2h_m + 2q_m)e^{-m_1}$ it holds that*

$$\sum_{i \in n', z_i' \geq 0} w_i z_i' \geq (1 - 6\varepsilon)G^+(X\beta).$$

Here the constant before the $\varepsilon$ increases for the following reason: assume that for some $z_i > 0$ we have $\|B_i\|_1 \geq (1 - 3\varepsilon)z_i$ then it holds that $\|B_i\|_1^2 \geq (1 - 6\varepsilon)z_i^2$.

**Dilation bounds** Here we have to cope with the additional factor of $\sqrt{n}$. Recall that if we choose $M_i$ solving the equation $q_{(M_{i-1},N)}(3) = q_{(M_i,N)}(2)$ then it holds that

$$q_{(i+2)}(1) - q_i(4) = q_{i+1}(3) - q_{i+1}(2) - (q_{i+2}(2) - q_{i+2}(1)) - (q_i(4) - q_i(3)).$$

We now have

$$q_{i+1}(3) - q_{i+1}(2) = \log_2\left(\frac{N\varepsilon^5}{32\sqrt{n}m_1\mu q_m}\right)$$

and

$$(q_{i+2}(2) - q_{i+2}(1)) + (q_i(4) - q_i(3)) = \log_2\left(\frac{64m_1 \ln(Nh_m/\delta)q_m h_m}{\sqrt{n}\varepsilon^9\delta}\right).$$

Further there is a change in LemmaC.10 as we have to deal with possible overhead coming from the square function. We set $R = \{i | 2^{-q_h(1)} > z_i > 2^{-q_h(4)}\}$ to be the set of relevant (positive) elements and $W_R = \{z_i \mid i \in R\}$.

**Lemma E.8.** *If $\sum_{i=1}^n z_i \leq 0$ and for all $i \leq h_m - 1$ it holds that $q_{(M_i N_i)}(4) < q_{(M_{i+k}N_{i+k})}(1)$ and $N_0 \geq N_0'$, then the expected contribution of any weight class $Y_1'$ is at most $k \cdot \|W_R\|_2^2$.*

*Proof.* Fix a level $h$ and a bucket $B$ at level $h$. Recall that $\sum_{i \in R} z_i \leq A_2 = A - A_1 = \sum_{i, z_i \geq 2^{-q_h(4)}} z_i$. Note that by Lemma E.3 we have that $\sum_{i \in Y_1' \cap B} \leq \frac{p_h(-A_2)}{N}$. Let $Z_i$ be the random variable where $Z_i = z_i$ if $i \in R$ is assigned to $B$ and 0 otherwise. Then the expected value of $Z = \max\{0, \sum_{i=1}^n Z_i\}$ is $\frac{p_h}{N} \cdot A_2$. Thus it holds that

$$\mathbb{E}(\max\{G(B), 0\}^2) \leq \mathbb{E}\left(\max\{Z - \frac{p_h(-A_2)}{N}\}^2\right) \leq \mathbb{E}\left(\max\{Z - \mathbb{E}(Z)\}^2\right)$$
$$= \text{Var}(Z) \leq \|W_R\|_2^2.$$

$\square$

Lemma C.11 and Lemma C.12 can be adapted as follows:

**Lemma E.9.** *If we choose $N_i = N := \max\{N_0^{(2)}, \frac{\sqrt{32q_m\mu}m_1 n^{0.75}}{\varepsilon^{2.5}}\}$ for all $i \in [h_m]$ and $M_i$ solving the equation $q_{(M_{i-1},N)}(3) = q_{(M_i,N)}(2)$ then the expected contribution of any weight class $W_q$ is at most $2\|W_q\|_1$.*

*Proof.* The proof uses a different idea as before: since $N$ is large enough, we only need 2 levels. More precisely we want to achieve $M_2 = N$. By our choice of $M_2$ this means

$$\log_2(\frac{8q_m\mu m_1 n}{\varepsilon^3 N}) = q_{(N,N)}(2) = q_{(n,N)}(3) = \log_2(\frac{Nn\varepsilon^2}{4n\sqrt{n}})$$

or equivalently

$$N = \sqrt{\frac{32 q_m \mu m_1 n^{1.5}}{\varepsilon^5}} = \frac{\sqrt{32 q_m \mu} m_1 n^{0.75}}{\varepsilon^{2.5}}.$$

$\square$

**Lemma E.10.** *If for some $k \in \mathbb{N}$ we choose $N_i = N \geq \frac{32 m_1 \mu q_m}{\varepsilon^5} \cdot \left( \frac{64 m_1^2 \ln(n) q_m h_m \sqrt{n}}{\varepsilon^9 \delta} \right)^{1/(k-1)}$ for all $i \in [h_m]$ and $M_i$ solving the equation $q_{(M_{i-1}, N)}(3) = q_{(M_i, N)}(2)$ then the expected contribution of any weight class $W_q$ is at most $k \|W_q\|_1$.*

The proof is the same as for Lemma C.12.

**Net Argument**

**Lemma E.11.** *For any $v, v' \in \mathbb{R}^n$ with $\|v - v'\|_1 \leq \varepsilon$ it holds that $|f_2(v) - f_2(v')| \leq (f_1(v) + \varepsilon)\varepsilon$.*

*Proof.* We have that $(\ell^2)'(v) = \frac{e^v}{e^v + 1} \cdot \ell(v) \leq \ell(v)$. Further since $\ell'(v) \leq 1$ we have that for any $\nu \in [0, 1]$ it holds that $|\ell(v + \nu(v' - v)) - \ell(v)| \leq (\ell(v) + \varepsilon)\varepsilon$. Thus we get that

$$|f_2(v) - f_2(v')| \leq \frac{1}{n} \cdot \sum_{i=1}^n |\ell(v_i)^2 - \ell(v_i')^2| \leq \frac{1}{n} \cdot \sum_{i=1}^n (\ell(v) + \varepsilon)\varepsilon = (f_1(v) + \varepsilon)\varepsilon$$

which proves the lemma. $\square$

**Lemma E.12.** *Assume that for $\beta \in \mathbb{R}^d$ it holds that $|f_2(X'\beta) - f_2(X\beta)| \leq \varepsilon$. Then for any $\beta' \in \mathbb{R}^d$ with $\|X\beta - X\beta'\|_1 \leq \varepsilon/(b^{h_m} h_m)$ it holds that $|f_2(X\beta') - f_2(X'\beta')| \leq \varepsilon + 2(f_1(X\beta') + \varepsilon)\varepsilon$.*

*Proof.* It holds that $\|X'(\beta - \beta')\|_1 = \|SX(\beta - \beta')\|_1 \leq b^{h_m} h_m \|X(\beta - \beta')\|_1 \leq \varepsilon$ since for each $i \in [n]$ there are at most $h_m$ columns $j$ such that $S_{ij} \neq 0$ and each entry of $S$ is bounded by $b^{h_m}$. Thus, by the triangle inequality and applying Lemma E.11 yields

$$|f_2(X\beta') - f_2(X'\beta')| \leq |f_2(X'\beta') - f_2(X'\beta)| + |f_2(X'\beta) - f_2(X\beta)| + |f_2(X\beta) - f_2(X\beta')|$$
$$\leq (f_1(X\beta') + \varepsilon)\varepsilon b^{-h_m} + \varepsilon + (f_1(X\beta') + \varepsilon)\varepsilon \leq \varepsilon + 2(f_1(X\beta') + \varepsilon)\varepsilon.$$

$\square$

Combining Lemma C.16 and Lemma E.12 we get:

**Lemma E.13.** *There exists a net $\mathcal{N} \subset \mathbb{R}^d$ with $|\mathcal{N}| = \exp\left(\mathcal{O}(d \ln(n))\right)$ such that if $|f_1(X'\beta) - f_1(X\beta)| \leq \varepsilon$ holds for any $\beta \in \mathcal{N}$ then for any $\beta' \in \mathbb{R}^d$ with $\|X\beta'\|_1 \leq n\mu$ it holds that $|f_2(X'\beta') - f_2(X\beta')| \leq \varepsilon(f_2(X\beta') + f_1(X\beta'))$.*

**Proof of Theorem 3** The proof of Theorem 3 works as the proof of Theorem 2, replacing the old lemmas with the new ones.

### E.1 LOWER BOUND

We note that the increased sketching dimension in terms of $\sqrt{n}$ comes from the inter norm inequality $\|x\|_1 \leq \sqrt{n} \|x\|_2$ and from more subtle details of the sketch. Lemma E.14 shows that there is no way to get around a factor of $\sqrt{n}$ using the CountMin-sketch. The proof gives an example where $\sqrt{n}$ is attained even for obtaining a superconstant (in $\mu$) approximation. It does not rule out the existence of some other method that allows a lower sketching dimension. For example Count-sketch is known to work for $\ell_1$ and $\ell_2$ norms simultaneously within polylogarithmic size (Clarkson & Woodruff, 2015a). But we stress that the standard sketches from the literature do not work for asymmetric functions since they confuse the signs of contributions leading to unbounded errors for our objective function or even for plain logistic regression, see (Munteanu et al., 2021).

**Lemma E.14.** *There exists a $\mu$-complex data example $X$ where our sketch with $o(\sqrt{n})$ rows fails to approximate $f$. Specifically, if $\lambda = 1$ it holds for the optimizer $\tilde{\beta} \in \arg\min_{\beta \in \mathbb{R}^d} f(SX\beta)$ that $f(X\tilde{\beta}) = \omega(\ln(\mu)^2) \cdot \min_{\beta \in \mathbb{R}^d} f(X\beta)$.*

*Proof.* Fix $\mu > 10$ and consider the following data

$$x_0 = (\sqrt{n})$$

$$x_i = (-1) \text{ for } i \in \left[1, n - \frac{n}{\mu}\right]$$

$$x_i = (1) \text{ for } i > n - \frac{n}{\mu}$$

As the example is 1-dimensional we only need to check the ratio for $\beta = 1$ and $\beta = -1$ in order to compute $\mu$ as multiplying with a scalar does dot not change the ratio between the sum of all positive points and the sum of all negative points. Also note that the ratio is inverted for $\beta = -1$ thus if the ratio is positive for $\beta = 1$ we do not need to check it for $\beta = -1$. Note that for $\beta = 1$ and $z = X\beta$ it holds that $\sum_{z_i > 0} z_i = \sqrt{n} + \frac{n}{\mu}$ and $\sum_{z_i < 0} |z_i| = n(1 - \frac{1}{\mu}) \geq \sqrt{n} + \frac{n}{\mu}$ if $n$ is sufficiently large. We thus have have

$$\mu_1(X) = \frac{n\left(1 - \frac{1}{\mu}\right)}{\sqrt{n} + \frac{n}{\mu}} \leq \frac{n}{\frac{n}{\mu}} = \mu$$

Further we have that $\sum_{z_i > 0} z_i^2 = n + \frac{n}{\mu} \leq 2n$ and $\sum_{z_i < 0} |z_i| = n(1 - \frac{1}{\mu}) \approx n$. Consequently we get that

$$\mu_2(X) = \frac{n + \frac{n}{\mu}}{n\left(1 - \frac{1}{\mu}\right)} \leq 2 < \mu$$

Since $d = 1$ this proves that our our example is $2\mu$-complex. Note that the following four facts hold for any level $h$:

- If for some $c$ we have that $p_h \leq 1/b$ then with probability $1/b$ row $x_0$ is not sampled at level $h$. In particular this implies that $x_0$ is only present at level 0 with high probability, i.e. probability at least $\sum_{h=1}^{h_m} p_h \leq \frac{2}{b}$;

- If $x_0$ is in a bucket with $3\sqrt{n} \geq 2\sqrt{n}/(1 - \frac{2}{\mu})$ elements then with high probability $G(B_0) \leq 0$;

- If $\frac{N_h}{n} \ll p_h \ll 1$ then with high probability $G(B) < 0$ for any bucket at level $h$ since the $\frac{\mu-1}{\mu} \cdot n \gg \frac{n}{\mu}$ negative elements cancel all positive rows;

- If $h = h_m$ then roughly $\frac{\mu-1}{\mu} \cdot N_u$ are $-1$ and $\frac{N_u}{\mu}$ are 1.

All of these follow from the Chernoff bounds using Lemma A.3. Thus if $N_0 \ll \sqrt{n}/3$ then $X' = SX$ mimics the instance $X \setminus \{x_0\}$, i.e. the instance $X$ with point $x_0$ removed, as $x_0$ is only appearing at level 0 where it is canceled by the other points. More precisely $X'$ consists of roughly $n' - \frac{n'}{\mu}$ copies of the point $-1$ and $\frac{n'}{\mu}$ copies of the point 1. After multiplying with the weights we are back to roughly $n - \frac{n}{\mu}$ times the point $-1$ and $\frac{n}{\mu}$ times the point 1. To keep the presentation simple we only consider the instance $X' = -(X \setminus \{x_0\})$. The proof works the same for other sketched instances that we obtain using the above facts. Consider the function

$$nf_1(X'r) = (n - \frac{n}{\mu}) \cdot \ell(-r) + \frac{n}{\mu} \cdot \ell(r) = n \cdot \ell(-r) + \frac{nr}{\mu}.$$

Thus, we have $f_1(X'r) = \ell(-r) + \frac{r}{\mu}$. Using that $\ell(r) < r + 1$ for all $r > 0$ we get

$$nf(X'r) = nf_1(X'r) + \sum_{i=1}^{n}(x_i - f_1(X'r))^2$$

$$\leq n \cdot \ell(-r) + \frac{nr}{\mu} + \frac{n(r+1)^2}{\mu} + (n - \frac{n}{\mu}) \cdot \ell(-2r)$$

$$\leq 2n \cdot \ell(-r) + \frac{nr}{\mu} + \frac{n(r+1)^2}{\mu}.$$

Using that $\ell(-r) \le e^{-r}$ it holds that

$$f(X'r) \le 2e^{-r} + \frac{r}{\mu} + \frac{(r+1)^2}{\mu}.$$

Taking the derivative we get

$$f'(X'r) \le -2e^{-r} + \frac{1}{\mu} + \frac{2(r+1)}{\mu}.$$

which is 0 if and only if $r = -\ln(\frac{2r+3}{\mu}) + \ln(2) = \Omega(\ln(\mu))$. This implies that for $\tilde{r} = \operatorname{argmin}_{r \in \mathbb{R}} f(Xr)$ we have that $\tilde{r} = \Omega(\ln(\mu))$. Now consider our original loss function $f(X\tilde{r})$. Here we have that

$$nf(Xr) = nf_1(Xr) + \sum_{i=1}^{n}(x_i - f_1(Xr))^2 \ge n \cdot \ell(-r) + \frac{nr}{\mu} + (\sqrt{n} \cdot r)^2/2$$

$$\ge n \cdot r^2/2.$$

In particular we have that $f(X\tilde{r}) = \Omega(\ln(\mu)^2)$. However for $r^*$ minimizing $f(Xr)$ we have that $nf(Xr) \le f(0) = \ln(2) = O(1)$. $\qquad\square$

# F    ADDITIONAL EXPERIMENTAL RESULTS

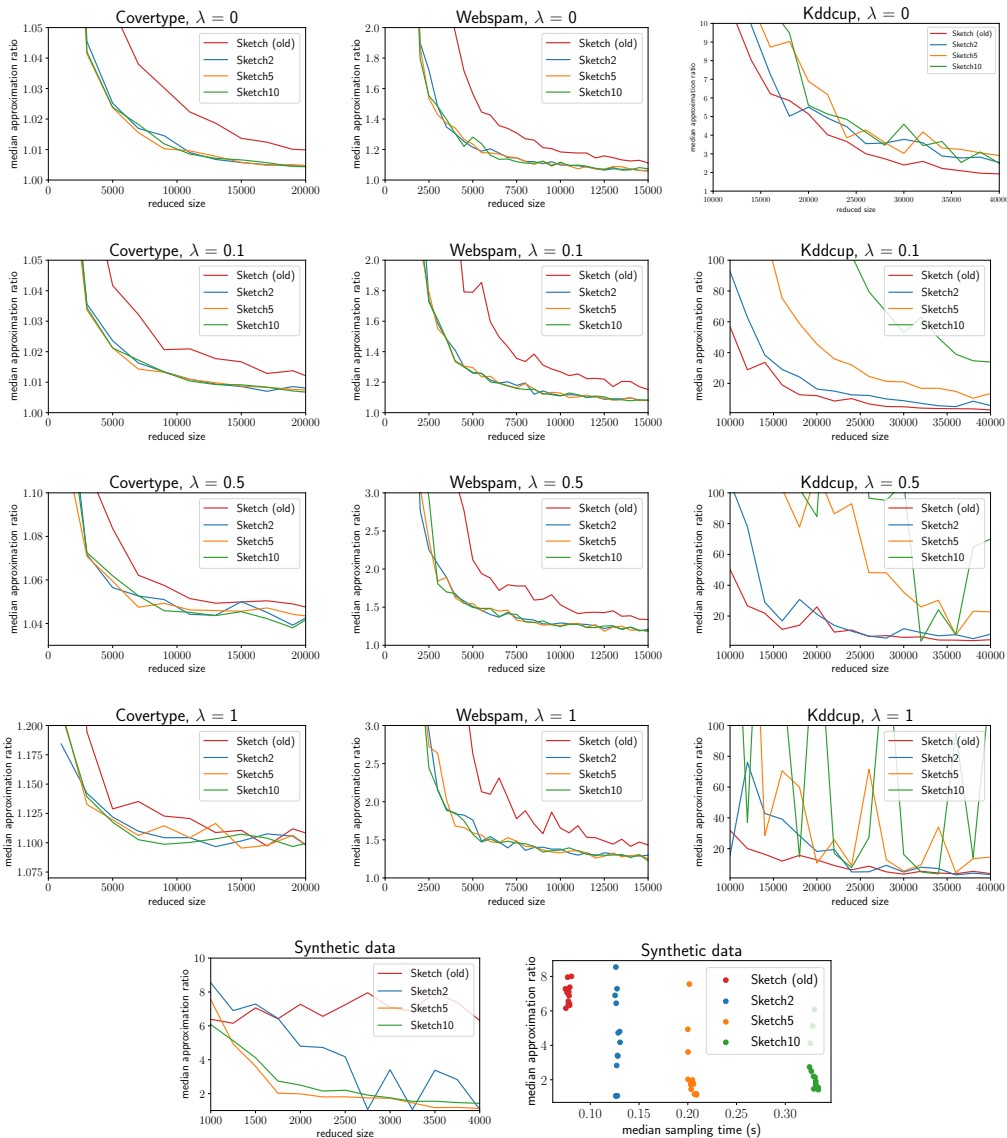

Figure 3: Comparison of the median approximation ratios of the old sketch versus the new sketch with various settings for the sparsity $s \in \{2, 5, 10\}$ as well as for the regularization parameter $\lambda \in \{0, .1, .5, 1\}$ for different real-world benchmark data (top, middle). Comparison of median approximation ratios (bottom left) and sketching times (bottom right) for our synthetic data.

## G  ADDITIONAL MATERIAL

**Comment on stochastic gradient descent (SGD) and supporting experiments** SGD does not work in the turnstile data stream setting (where positive and negative updates are allowed), and is inherently sequential. In contrast, oblivious linear sketching allows for simple handling of turnstile data streams and distributed or parallel computations, which we motivate in our paper.

Our bounds are *multiplicative* error guarantees that are relative to the optimal loss $f(X\beta_{OPT})$. Known regret or generalization results for SGD bound the probability of misclassification $P(y_ix_i\beta < 0) < \varepsilon$, which allows to *ignore* a few highly important (and expensive) points and give an *additive* error on the loss $|f(X\beta_{SGD}) - f(X\beta_{OPT})| \leq B$. Here $B$ depends on properties of $f$, $X$, and on the distance of an initial guess $\beta_0$ to the optimal solution $\|\beta_0 - \beta_{OPT}\|$. Thus $B$ cannot be charged uniformly for the optimal loss, and instead can be arbitrarily large.

The reason SGD and online gradient descent do not work in our setting is that they miss (in most iterations) highly important points when there are only few of them (this is also the issue with uniform sampling). This was pointed out in previous related work, e.g., (Munteanu et al., 2018, Section C) and (Munteanu et al., 2021, Section 6), who constructed synthetic data with only 2 out of $n$ such important points and additionally demonstrated empirically how bad SGD can perform even on mild data with $\mu = 1$.

Below we add SGD to our empirical results on real world data. SGD performs quite well, though not better than sketching. The reported performance of SGD is the median approximation ratio over 21 independent repetitions of *one full pass* over the data, to be a baseline comparable to the sketches. We note that plotting the iteration-wise error would make the results for SGD look much worse.

For our synthetic data (described in detail below), instead of 2 out of $n$ heavy points (as in previous work), we have $\Theta(d)$ out of $n$ heavy points. Since SGD misses those in most batches, the instance looks separable to SGD in almost all iterations, although the original instance is inseparable. This results in approximation ratios around $15\,000$ (note the logarithmic scale on the vertical axis). In contrast, our sketch and the previous sketch give small constant approximations.

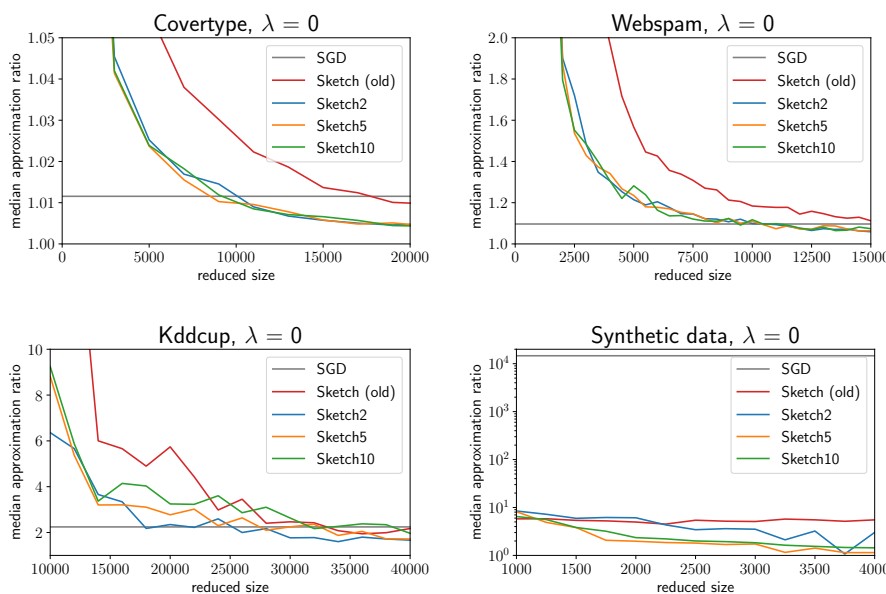

Figure 4: Median approximation ratios for plain logistic regression ($\lambda = 0$). SGD is compared to the old sketch as well as the new sketch with various settings for the sparsity $s \in \{2, 5, 10\}$ on different real-world benchmark data, and on our synthetic data.

**Added experiments for $\ell_1$-regression** We implemented the Cauchy sketch (Indyk, 2006; Sohler & Woodruff, 2011; Woodruff, 2021) that simply consists of i.i.d. standard Cauchy entries. The sketching matrix is then multiplied by the data matrix and the sketched $\ell_1$ regression problem is solved. The plots show the median approximation ratio over 21 repetitions for each target size of the sketch. We see that the new sketch, using any degree of sparsity $s \in \{2, 5, 10\}$, outperforms the Cauchy sketch by a large margin in terms of approximation factor (while being a lot faster to apply than the dense matrix multiplication).

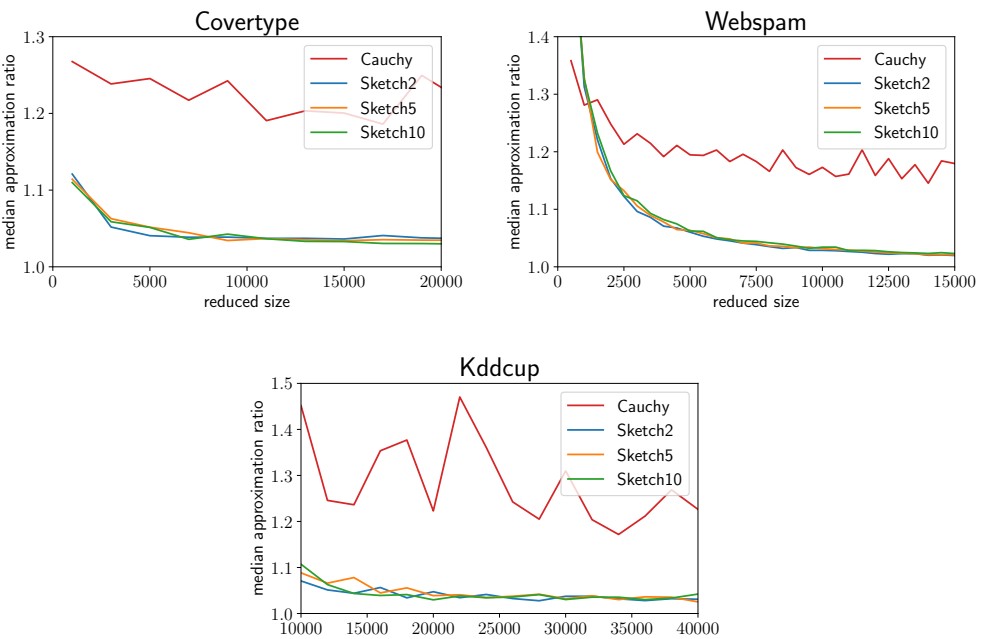

Figure 5: Comparison of the median approximation ratios for $\ell_1$-regression of the Cauchy sketch versus our new sketch with various settings for the sparsity $s \in \{2, 5, 10\}$ for different real-world benchmark data.

**Synthetic data set** The synthetic data set consists of $2n = 40\,000$ data points. The dimension of the data points is $d = 100$.

- There are $(n - n/10 - 2d)$ points of the form $(-1, -1, ..., -1)$. The optimization of $\beta$ focuses on these points.

- $(n/10)$ points are of the form $(1, 1, ..., 1)$. These points are only added to obtain clean plots. If they are omitted, then the gap between the optimal $\beta$ on the original data and the optimal $\beta$ on a bad sketch or uniform sample becomes worse, and "wiggly".

- $d$ points are of the form $(-n, -n, \ldots, -n)$. They are oriented in the same direction as the first set of points. They will mostly be ignored in the optimization since there are only a few of them, but they can cancel the following heavy hitters.

- For each $i \in [d]$, we add one vector of the form $(n \cdot e_i)$, where $e_i$ is $i$th standard basis vector. These points are the *heavy hitters* pointing away from most other points. For any good relative error sketch, it is crucial to preserve all of them;

- We add $n$ times the point $(0, 0, \ldots, 0)$. These points are needed to ensure that the instance works in a labeled setting (to be a natural data set for logistic regression).

- The labels of all points unequal to the all zero vector are set to $1$. All zero vectors are assigned the label $-1$.

The idea behind this instance is as follows: if the sketch maps any point of the form $(n \cdot e_i)$ into the same bucket as a $(-n, -n, \ldots, -n)$-vector, then the instance will become almost separable, so the sketch will have a cheap solution, meaning that there exists some $\beta$ such that the logistic loss on the sketch is low. However, on the original instance, the logistic loss of the same $\beta$ will be large due to the loss associated with $(n \cdot e_i)\beta$. This implies a large approximation ratio. For small sketch sizes, the old sketch has a relatively high probability for this bad event to happen, when hashing each point into a single bucket. Our new sketch will likely preserve all points of the form $(n \cdot e_i)$ on *most* of the multiple sub levels. This preserves the cost even if our sketch size is small (almost linear).

**Pseudocode of our sketching algorithm** Algorithm 1 implements the first step of the sketch & solve paradigm for approximating logistic or $\ell_1$ regression. The changes in comparison with (Munteanu et al., 2021) are highlighted in red.

---

**Algorithm 1** Oblivious sketching algorithm for logistic regression.

---

**Input:** Data $X \in \mathbb{R}^{n \times d}$, number of rows $k = N \cdot h_m + N_u$, parameters $b > 1, s \geq 1$ where $N = s \cdot N'$ for some $N' \in \mathbb{N}$;
**Output:** weighted Sketch $C = (X', w) \in \mathbb{R}^{k \times d}$ with $k$ rows.;
1: **for** $h = 0 \ldots h_m$ **do**                                     ▷ construct levels $0, \ldots h_m$ of the sketch
2:     initialize sketch $X'_h = \mathbf{0} \in \mathbb{R}^{N \times d}$ at level $h$;
3:     initialize weights $w_h = b^h \cdot \mathbf{1} \in \mathbb{R}^N$ at level $h$;
4: set $w_0 = \frac{w_0}{s}$;                                     ▷ adapt weights on level 0 to sparsity $s$
5: **for** $i = 1 \ldots n$ **do**                                     ▷ sketch the data
6:     **for** $l = 1 \ldots s$ **do**                                     ▷ densify level 0
7:         draw a random number $B_i \in [N']$;
8:         add $x_i$ to the $((l-1) \cdot N' + B_i)$-th row of $X'_0$;
9:     assign $x_i$ to level $h \in [1, h_m - 1]$ with probability $p_h = \frac{1}{b^h}$;
10:     draw a random number $B_i \in [N]$;
11:     add $x_i$ to the $B_i$-th row of $X'_h$;
12:     add $x_i$ to uniform sampling level $h_m$ with probability $p_{h_m} = \frac{1}{b^{h_m}}$;
13: Set $X' = (X'_0, X'_1, \ldots X'_{h_m})$;
14: Set $w = (w_0, w_1, \ldots w_{h_m})$;
15: **return** $C = (X', w)$;

---

