# OpenReview forum: "Almost Linear Constant-Factor Sketching for $\ell_1$ and Logistic Regression"
_ICLR.cc/2023/Conference — ICLR 2023 poster_

### Official Review · Reviewer_mC84 · 2022-10-25

**Confidence:** 3
**Correctness:** 4
**Technical Novelty And Significance:** 4
**Empirical Novelty And Significance:** 3
**Recommendation:** 8

**Clarity, Quality, Novelty And Reproducibility:**

It would perhaps be more useful to describe the entire process of sketch and solve as an algorithm in place of describing it in a paragraph. There does not seem to be any empirical validation for the $\ell_1$- regression problem. I could not find details of how synthetic data is generated.

Although sketches have been applied to logistic and $\ell_1$ regression in earlier works, the improvement obtained here is vast. I did not check the proofs, but the technical ideas given in the main paper appear convincing.

**Strength And Weaknesses:**

Strengths:
1. Vast improvements in oblivious sketching dimension for logistic regression compared to previous works.
2.  Sketch for variance regularized logistic regression
3. Empirical evaluations

Weakness:

The only weakness I feel is that the paper may be difficult to read for those who are unfamiliar with the related works. There are a few points I mention in the section below which might help improving the clarity

**Summary Of The Paper:**

The paper presents oblivious sketches for logistic regression with significantly improved sketching sizes both in terms of $d$- the dimension of the data and $\mu$ the measure of complexity of compressing the data. The paper also gives sketches for the $\ell_1$ regression and variance regularized logistic regression. Theoretical claims are supported by empirical evaluations on both real and synthetic data.

**Summary Of The Review:**

Overall, the paper is more of theoretical nature and will be of great interest to the researchers working in the "sketch and solve" paradigm. Also, logistic regression being a very important problem in machine learning, the paper would be of interest to broader community too. I recommend an accept

---

> ### Author Response · Authors · 2022-11-14
> **Thank you for the review**
>
> Thank you for appreciating our work and for giving explicit hints on how to improve the clarity of our paper. We will incorporate your comments into the main paper as best as possible given the page limitations. For your reference during the rebuttal, we have already incorporated your comments in Appendix G as follows:
>
> 1) We have added an explanation with pseudocode for the sketch & solve paradigm, as well as pseudocode for our specific sketching algorithm.
>
> 2) We have added empirical results for $\ell_1$ regression on real world benchmark data. Here, we implemented the Cauchy sketch and compared our new Sketch$s$, for $s\in$ { $2,5,10$ } (similar to the case of logistic regression). The results are in favor of our new sketch.
>
> 3) We have added a description of the synthetic data set together with intuition why this is a complicated instance for the old sketch, while our new sketch can handle it much better.

---

### Official Review · Reviewer_bsu6 · 2022-10-25

**Confidence:** 3
**Correctness:** 4
**Technical Novelty And Significance:** 3
**Empirical Novelty And Significance:** 3
**Recommendation:** 8

**Clarity, Quality, Novelty And Reproducibility:**

Quality and Clarity: The paper is well-written and clear.
Novelty: The paper introduces a new analysis for an existing algorithm, slightly modified.
Reproducibility: The authors point the reader to prior work for code that can be modified to run the experiments.

**Strength And Weaknesses:**

Strengths:
1. This is an interesting paper with a clear improvement over prior work.
2. The analysis involves several new ideas.


**Summary Of The Paper:**

This paper improves upon previous oblivious sketching and turnstile streaming results for $\ell_1$ and logistic regression, achieving constant factor approximation, and an efficient algorithm in the sketched space. They demonstrate:

1. Based on a modification of Munteanu et al. (2021) (with significant change of the analysis), they are able to achieve an $O(1)$-factor approximation algorithm with sketch size $\tilde{O}(\mu d^{1+c})$ for any $c > 0$, for the problem of logistic regression. Here $\mu$ is a data-dependent parameter which captures the complexity of compressing the data for logistic regression.

2. To obtain a $1+\epsilon$ guarantee, they demonstrate that a sketch size of $(\mu d \log(n)/\epsilon)^{O(1/\epsilon)}$ is sufficient.

3. They show that their sketch can also approximate variance-based regularized logistic regression
within an $O(1)$ factor if the dependence on $n$ in the sketching dimension is increased to $n^{0.5+c}$ for
any $c > 0$.

The authors then experimentally demonstrate the performance of their algorithms in comparison to prior work, with performance usually better, and just a little worse than prior work on one benchmark dataset.





**Summary Of The Review:**

I think this is a good paper and vote to accept it. The paper produces a clear improvement over prior work, building off the work while complementing it with a significant change of analysis.

---

> ### Author Response · Authors · 2022-11-14
> **Thank you for the review**
>
> Thank you for appreciating our work. Regarding your comment on reproducibility: "The authors point the reader to prior work for code that can be modified to run the experiments", we will release the readily modified code and data in a public git repository so as to ensure reproducibility of all experimental results in our paper, and to provide the code to the community for further use.

---

### Official Review · Reviewer_yfhf · 2022-10-31

**Confidence:** 3
**Clarity, Quality, Novelty And Reproducibility:** Most of this is discussed in the weak…
**Correctness:** 4
**Technical Novelty And Significance:** 3
**Empirical Novelty And Significance:** 2
**Recommendation:** 5

**Strength And Weaknesses:**

Strengths:

The proposed algorithm is a simple modification of the previous approach, but comes with much stronger guarantees and achieves better results in practice.

Weaknesses:

My main concern is that the paper is not written in a way such that it can really be self-sufficient without reading and understanding Munteanu et al.'21 and related literature. This makes the contribution harder to evaluate and makes it appear to be not as significant. There are several issues in the writing and discussion of the setup:

1. To begin with, a much better job needs to be done to motivate the problem. Since the goal is to solve the logistic regression problem without storing all the datapoints, why is SGD/online gradient descent not the right approach? It would only require memory linear in dimension, even better than the proposed approach. It comes with regret guarantees too, are those sub-optimal compared to the approximation ratios which the paper obtains?

2. Various approaches have been proposed to sketch solutions to linear classification problems, such as "Linear and Kernel Classification in the Streaming Model: Improved Bounds for Heavy Hitters", "Sketching Linear Classifiers over Data Streams", "Random Projections for Classification: A Recovery Approach", "Finding needles in compressed haystacks", "Compressed Classification from Learned Measurements", how do the sketches in these paper compare with the proposed approach? It appears to me that these papers compress the dimensionality d of the datapoints, whereas the current paper compresses the number of datapoints n, but some more discussion of the pros and cons would be useful to situate the work better.

3. The discussion of related work is often that as well organized. For e.g. Sec 1.2 on L1 regression combines various discussions of the current paper with previous approaches in a way which makes it a bit hard to follow the thread.

4. The paper would benefit from an algorithm box with a detailed description of the algorithm (perhaps also pointing out the part where the algorithm differs from Munteanu et al.'21).

5. Similarly, it is nice that the authors included Section 3 on the proof overview, but I found it hard to follow without first going through Munteanu et al.'21.

**Summary Of The Paper:**

The paper considers the problem of using sketching techniques for logistic regression. It builds on the previous work of Munteanu et al.'21 which developed an approach to sketch n datapoints in d dimensions to poly(\mu,d,log n) weighted datapoints (in d dimensions) such that solving the logistic regression problem on the smaller dataset gives a log(n) approximation to the function value obtained by the solution on the original dataset. Here, \mu is a data-dependent parameter which captures the complexity of sketching. Though Munteanu et al.'21 obtain a poly(\mu,d,log n) sized sketch, the polynomial dependence on d is not optimal. The current paper closes this gap and shows that a sketching dimension only slightly superlinear in d suffices to obtain a constant approximation to the logistic regression problem. Similar approximation results are also obtained for the L1 loss. The improvement comes from a small modification in the algorithm, and a much tighter analysis. The modification in the algorithm is to hash each element to multiple buckets instead of a single bucket in the CountMin Sketch.

The paper validates the algorithm via similar experiments as in Munteanu et al.'21. The proposed algorithm generally performs better than the previous approach in the experiments.



**Summary Of The Review:**

In summary, I think this paper has a good contribution and achieves tight bounds for a natural problem. But especially for a venue like ICLR where much of the audience may be unfamiliar with the developments for this problem, it is important for the paper to be written and presented such that familiarity with prior work is not as essential. The paper falls short on this regard, and I would place it below the acceptance threshold for this reason.

---

> ### Author Response · Authors · 2022-11-14
> **Thank you for the review**
>
> Thank you for your thorough review. We will incorporate your concerns into the main paper as best as we can given the page limitations. For your reference during the rebuttal, we have addressed your comments in Appendix G as follows:
>
> 1) The goal is to give a low memory oblivious linear sketch that allows sketching the data in a turnstile data stream as well as in distributed (or parallel) settings, please see the introduction. These settings are simple to handle in our 'sketch & solve' setting but are not supported by SGD. Further it is known (cf. Munteanu et al. 2018, 2021) that SGD regret bounds give only arbitrarily large additive errors for the problem studied in our paper, rather than the multiplicative and uniformly bounded error bounds as we have. We will add these arguments together with references early in the introduction. Further, we will point to the detailed explanations together with empirical illustrations, currently in Appendix G.
>
> 2) Thank you for pointing us to those references. The papers consider very different problems from the one studied in our paper:
>     - As you already mentioned, they aim at reducing the dimensionality of input points rather than reducing the number of input points. Both goals are important in different settings but our focus is explicitly on reducing $n$ in the case $n \gg d$.
>     - Another important difference is that all of the references have some sort of $\ell_2$ regularization. On the one hand, this simplifies logistic loss to an almost symmetric function, sketchable by means of standard techniques such as CountSketch and Johnson-Lindenstrauss embeddings (which fail in our unrestricted setting). On the other hand, we mentioned in the paper that adding this sort of regularization simplifies our problem so much that it becomes readily solvable by uniform sampling (Samadian et al., 2020).
>     - Two of the given references focus on sketching online gradient descent, which, as we have already argued and demonstrated, fails for our problem (see the above as well as Appendix G).
>     - One paper explicitly studies SVM, which is a completely different problem than logistic regression (see below). Another paper is on neural networks learning a dimensionality reduction before classification, with a mixture of Frobenius norm error (again symmetric) and cross-entropy, which we consider unrelated to our work.
>     - We stress that the problem that we study is not "sketching classification or SVMs", it is specifically "sketching logistic regression" (and $\ell_1$ regression) that we focus on. Logistic regression is often used as a linear classifier, but is not limited to classification. Its main purpose is a regression model for estimating Bernoulli probabilities by a logistic distribution based on a linear predictor $x^T\beta$ (McCullagh & Nelder, 1989). This being said, our aim is to preserve the loss function (negative log-likelihood) of the probabilistic model of logistic regression.
>
> 3) We are sorry that the discussion of related work may not be organized well enough. In particular, there are a lot of related but very different results on $\ell_1$-regression,  complicating the presentation in limited space. We will give an improved concise presentation in Section 1.2, and refer to an appendix when necessary to give more detail.
>
> 4) We have added an explanation with pseudocode for the sketch & solve paradigm, as well as pseudocode for our specific sketching algorithm, highlighting the differences to the sketch of (Munteanu et al. 2021).
>
> 5) Given the page limitations, there is a trade-off between describing the contents of the paper (Munteanu et al. 2021) and describing the modifications and novel arguments to achieve our new bounds, i.e., our main contributions. We will improve this and make the presentation as self-contained as possible in the next revision.

---

### Decision · Program_Chairs · 2023-01-20

**Decision:**

Accept: poster

**Justification For Why Not Higher Score:**

This is kind of a niche, if fundamental and reasonably well-motivated, problem.  At least so far, interest in this line of research is mainly from theoretical computer scientists, which make up a small fraction of ICLR attendees.  This paper’s results are a significant step forward, but arguably not a breakthrough.

**Justification For Why Not Lower Score:**

Please see the above.


**Metareview: Summary, Strengths And Weaknesses:**

The authors present new algorithms for solving logistic and ell-one regression problems through “sketching”, i.e. projecting the problem onto lower dimensions.  While their algorithm is a slight modification of earlier algorithms, they establish much stronger guarantees for it – substantial new ideas are required for this analysis.

This is a significant step forward in an interesting and important line of research.


**Note From Pc:**

if the above contains the word "oral" or "spotlight" please see: "oral" presentation means -> notable-top-5% and "spotlight" means -> notable-top-25%. As stated in our emails, we are disassociating presentation type from AC recommendations